# SADA: Safe and Adaptive Inference with Multiple Black-Box Predictions

## Abstract

Real-world applications often face scarce labeled data due to the high cost and time requirements of gold-standard experiments, whereas unlabeled data are typically abundant. With the growing adoption of machine learning techniques, it has become increasingly feasible to generate multiple predicted labels using a variety of models and algorithms, including deep learning, large language models, and generative AI. In this paper, we propose a novel approach that safely and adaptively aggregates multiple black-box predictions with unknown quality while preserving valid statistical inference. Our method provides two key guarantees: (i) it never performs worse than using the labeled data alone, regardless of the quality of the predictions; and (ii) if any one of the predictions (without knowing which one) perfectly fits the ground truth, the algorithm adaptively exploits this to achieve either a faster convergence rate or the semiparametric efficiency bound. We demonstrate the effectiveness of the proposed algorithm through experiments on both synthetic and benchmark datasets.

## 1 Introduction

In real-world applications, labeled data are often expensive, time-consuming, or requires expert knowledge to obtain, whereas unlabeled data are abundant and easier to collect. With the rapid advancement of machine learning and generative AI, generating predicted labels using powerful tools such as large language models (LLMs) and other algorithms has become easier than ever before. With predictions derived from a single source, many methods have been proposed, such as self-training (Lee, 2013; Zhu et al., 2023) and prediction-powered inference (PPI) (Angelopoulos et al., 2023; 2024), to obtain better predictions or inference results under a variety of conditions.

However, the availability of multiple predictions introduces practical challenges. Outputs from different models–such as GPT, Llama, or DeepSeek–often differ, sometimes substantial; and the quality of predictions from black-box models can be highly variable. In particular, low-quality or poorly calibrated predictions can introduce significant noise, increasing variance and leading to unreliable inference. While one can evaluate the quality of each prediction or identify which model's output is closest to the ground truth, a more compelling goal is to develop a principled method to aggregate multiple sources of predictions such that the resulting inference is guaranteed to perform no worse than using the labeled data alone. Moreover, if any one of the predictions is perfectly accurate or satisfies some ideal conditions, the aggregation method should achieve performance equivalent to using that prediction alone. Thus, this paper aims to answer the following question:

> *Given multiple predicted labels with unknown quality, how can we aggregate them in a safe and data-adaptive manner to improve estimation and inference*?

To answer this question, we introduce a novel method that can safely and adaptively aggregate predictions from multiple black-box models, enabling valid and more informative inference. The proposed method has two key highlights:

- **Safety:** The proposed method is guaranteed to perform no worse than the naive estimator (using the labeled data alone) in terms of mean squared error, regardless of the choice of machine learning models or their prediction accuracy. This means the method remains valid even when the ML predictions are arbitrarily misspecified.

- **Adaptivity:** With multiple predictions, the proposed method adaptively utilizes more information of the better predictions to reduce variance, while downweighting poor predictions to avoid variance inflation. In particular, when one prediction is perfectly accurate, the method can effectively pick it up and performs as if we knew the ground truth of the unlabeled data.

## 1.1 OUR NOVEL CONTRIBUTIONS

First, we consider a semi-supervised setting where multiple sets of predicted labels are available, without making any assumptions about their quality or requiring prior knowledge of which predictions are more accurate. The predicted labels are also not needed to share the same scale or format, either with each other or with the true labels. This bridges the gap between advanced machine learning tools and principled methods for leveraging them to improve the inference results.

Second, we propose a safe and adaptive approach for aggregating multiple black-box predictions. Our method assigns data-driven weights to the predictions, effectively leveraging helpful information to reduce bias while mitigating the influence of harmful information that could increase variance. Importantly, it is guaranteed to perform no worse than using the labeled data alone, regardless of the quality of black-box predictions.

Third, we demonstrate that if any one of the predictions (no need to know which one) is perfectly accurate, the proposed estimator achieves a faster convergence rate, same as the oracle estimator that knows the ground truth of the unlabeled data. In addition, if we restrict the predictions to be deterministic functions of the observed features, say, generated by some pre-trained learners, and if any one of the predictions (no need to know which one) satisfies some ideal conditions, then the proposed estimator achieves the best possible estimation efficiency among all regular asymptotically linear estimators; i.e., it attains the semiparametric efficiency bound.

## 1.2 RELATED WORK

**Semi-supervised learning (SSL)** SSL has become a prominent approach in machine learning and statistics, leveraging both limited labeled data and abundant unlabeled data to enhance the performance of models (Grandvalet & Bengio, 2004; Zhu, 2005; Pan & Yang, 2010).

Over the past two decades, a wide range of SSL algorithms has been proposed. These methods differ in the assumptions, in how they utilize unlabeled data, and in their relationship to supervised learning approaches (van Engelen & Hoos, 2020). Broadly, these algorithms can be categorized into two types: *inductive* and *transductive* methods. Inductive methods, similar to supervised learning, use a pre-trained model to assign labels to unlabeled data. Examples include self-training (Yarowsky, 1995; Lee, 2013; Berthelot et al., 2019; 2020), co-training (Blum & Mitchell, 1998; Wang & Zhou, 2010; Deng & Guo, 2011), pseudo-labeled boosting methods (Zhou, 2012), unsupervised preprocessing (Sheikhpour et al., 2017). In contrast, transductive methods do not produce a generalizable model; instead, they predict labels by directly propagating information through connections between data points. It typically defines a graph over all data points, both labeled and unlabeled, encoding the pairwise similarity of data points with possibly weighted edges (Jebara et al., 2009; Liu et al., 2012; Subramanya & Talukdar, 2014). Additionally, significant progress has been made in recent years to understand, as well as how to leverage, the statistical benefits of the unlabeled data. Chakrabortty & Cai (2018) and Azriel et al. (2022) studied linear regression problems within the SSL framework and proposed estimators that are more efficient than ordinary least squares (OLS) which relies solely on labeled data. Song et al. (2023) further extended this framework to general M-estimation problems. The methodology has also been adapted to high-dimensional settings, where the number of features exceeds the sample size (Zhang et al., 2019; Cai & Guo, 2020; Zhang & Bradic, 2022; Deng et al., 2024). Applications of SSL have expanded beyond statistical models to include both 2D computer vision tasks (Jeong et al., 2019; Liu et al., 2021; Tang et al., 2021; Zhou et al., 2022) and 3D object detection problems (Wang et al., 2021; Park et al., 2022; Li et al., 2023; Liu et al., 2023).

**Prediction-powered inference (PPI)** In the past few years, a growing body of research has focused on enhancing statistical inference by incorporating predictions from black-box AI/ML models (Wang et al., 2020; Motwani & Witten, 2023). In particular, Angelopoulos et al. (2023) introduced *prediction-powered inference* (PPI), a framework that enables valid inference even when the predictive model is of low quality. However, PPI might perform worse in estimation efficiency compared to the

naive method that uses labeled data only. This limitation has motivated further research aimed at improving the efficiency of PPI or integrating it with ideas from other statistical and machine learning frameworks. Examples include PPI++ (Angelopoulos et al., 2024), cross PPI (Zrnic & Candès, 2024), stratified PPI (Fisch et al., 2024), and recalibrated PPI (Ji et al., 2025).

In related work, Zhu et al. (2023) proposed a doubly robust self-training method that achieves faster convergence rates when predictions are highly accurate. Miao et al. (2024) introduced a post-prediction adaptive inference approach that ensures valid statistical inference without relying on assumptions about the ML predictions. Gan et al. (2024) explored a broader class of imputed loss functions to enhance modeling flexibility and efficiency. Gronsbell et al. (2025) focused on inference under squared error loss, situating PPI within the broader context of semiparametric theory. Bartolomeis et al. (2025) introduced a framework that integrates the predictions from multiple foundation models with randomized experiments while preserving valid statistical inference.

**Missing data and causal inference**   SSL is also closely related to missing data and causal inference (Rubin, 1974; 1976). In those problems, common approaches include likelihood-based inference (Dempster et al., 1977; Ibrahim, 1990), imputation methods (Rubin & Schenker, 1986; Rubin, 2004; Vach & Schumacher, 1993), Bayesian approaches (Rubin, 1976), and semiparametric methods (Robins et al., 1994; Zhao et al., 1996). For more complex missing-not-at-random scenarios, earlier work established identification under specific modeling assumptions, such as outcome-selection models (Heckman, 1979), pattern-mixture parametrizations (Little, 1993; 1994), graphical models (Fay, 1986; Ma et al., 2003), and sensitivity analysis techniques (Rotnitzky et al., 1998; Robins et al., 2000).

## 2   PROBLEM SETUP

We first introduce some notations we use throughout. All vectors are assumed to be column vectors unless otherwise specified. Let $\mathbf{u}^{\otimes 2} = \mathbf{u}\mathbf{u}^{\mathrm{T}}$ for a vector $\mathbf{u}$. We denote $\mathbf{A} \preceq \mathbf{B}$ for two symmetric square matrices $\mathbf{A}$ and $\mathbf{B}$ if $\mathbf{B} - \mathbf{A}$ is positive semi-definite. We generally use capital letters to denote random variables, and the corresponding lowercase letters to denote their realizations. We use $\mathbb{P}$ to denote the probability measure and $\mathbb{E}$ to denote the expectation. For two random vectors $\mathbf{U}$ and $\mathbf{V}$, let $\mathrm{cov}(\mathbf{U}, \mathbf{V}) = \mathbb{E}[\{\mathbf{U} - \mathbb{E}(\mathbf{U})\}\{\mathbf{V} - \mathbb{E}(\mathbf{V})\}^{\mathrm{T}}]$ and $\mathrm{var}(\mathbf{U}) = \mathrm{cov}(\mathbf{U}, \mathbf{U})$.

**SSL, objective and the naive estimator**   We consider a standard SSL setting that consists of a set of $n$ labeled samples, $\mathcal{L} = \{(\mathbf{x}_i, y_i), i = 1, \ldots, n\}$ and a set of $N - n$ unlabeled samples, $\mathcal{U} = \{\mathbf{x}_i, i = n + 1, \ldots, N\}$, drawn from some underlying distribution of random variables $(\mathbf{X}, Y)$. Let $\ell_{\boldsymbol{\theta}}(\mathbf{x}, y)$ be a convex loss function. In this paper, we focus on a $p$-dimensional parameter, $\boldsymbol{\theta}^* \in \Theta \subset \mathbb{R}^p$, defined as

$$\boldsymbol{\theta}^* = \arg\min_{\boldsymbol{\theta} \in \Theta} \mathbb{E}\{\ell_{\boldsymbol{\theta}}(\mathbf{X}, Y)\}.$$

We define the score function $\mathbf{s}(\mathbf{x}, y; \boldsymbol{\theta}) = \partial \ell_{\boldsymbol{\theta}}(\mathbf{x}, y)/\partial\boldsymbol{\theta}$. Then, we can write $\boldsymbol{\theta}^*$ as the solution to the estimating equation

$$\mathbb{E}\{\mathbf{s}(\mathbf{X}, Y; \boldsymbol{\theta})\} = \mathbf{0}. \tag{1}$$

The definition of $\boldsymbol{\theta}^*$ is intentionally broad and includes many commonly studied parameters. For example, if the goal is to estimate the outcome mean $\mathbb{E}(Y)$, one may define the loss function as $\ell_{\theta}(\mathbf{x}, y) = (y - \theta)^2/2$, which yields the score function $\mathbf{s}(\mathbf{x}, y; \theta) = y - \theta$. As another example, when the parameter of interest is the coefficient in a linear regression model, the loss can be defined as $\ell_{\boldsymbol{\theta}}(\mathbf{x}, y) = (y - \mathbf{x}^{\mathrm{T}}\boldsymbol{\theta})^2/2$, leading to the score function $\mathbf{s}(\mathbf{x}, y; \boldsymbol{\theta}) = (y - \mathbf{x}^{\mathrm{T}}\boldsymbol{\theta})\mathbf{x}$.

The naive estimator, $\widehat{\boldsymbol{\theta}}^{\mathrm{nv}}$, which only uses the labeled data, solves

$$\frac{1}{n}\sum_{i=1}^{n} \mathbf{s}(\mathbf{x}_i, y_i; \boldsymbol{\theta}) = \mathbf{0}.$$

Assume the Hessian matrix $\mathbf{H} = \mathbb{E}\{\partial \mathbf{s}(\mathbf{X}, Y; \boldsymbol{\theta}^*)^{\mathrm{T}}/\partial\boldsymbol{\theta}\}$ exists and is nonsingular. Under some regularity conditions (see Assumption 1), the mean squared error of $\widehat{\boldsymbol{\theta}}^{\mathrm{nv}}$ equals

$$\mathbb{E}\{(\widehat{\boldsymbol{\theta}}^{\mathrm{nv}} - \boldsymbol{\theta}^*)^{\otimes 2}\} = \frac{1}{n}\mathbf{H}^{-1}\boldsymbol{\Sigma}_{\mathrm{nv}}\mathbf{H}^{-1}, \tag{2}$$

where $\boldsymbol{\Sigma}_{\mathrm{nv}} = \mathrm{var}\{\mathbf{s}(\mathbf{X}, Y; \boldsymbol{\theta}^*)\}$.

**Availability of multiple predicted labels**   We consider the availability of predicted labels from $K$ black-box models, denoted as $\widehat{\mathbf{y}}_i = (\widehat{y}_{1,i}, \ldots, \widehat{y}_{K,i})$, for each subject $i = 1, \ldots, N$. For example, in our application of wine reviews (see Section 5.2), a LLM can quickly predict a score based on information such as the wine's name, price, vineyard, and other attributes. Multiple LLMs–such as GPT, Llama, and DeepSeek–can be used to generate a range of predictions. We make no assumptions on the quality of these predictions. The data structure and generating process are illustrated in Table 1 and left panel of Figure 1.

Table 1: Data structure in SSL with multiple sets of predicted labels.

|  | Unit | Feature | Label | Multiple predicted labels |
|---|---|---|---|---|
| Labeled data $\mathcal{L}$ | 1 | $\mathbf{x}_1$ | $y_1$ | $\widehat{\mathbf{y}}_1 = (\widehat{y}_{1,1}, \ldots, \widehat{y}_{K,1})^{\mathrm{T}}$ |
|  | $\vdots$ | $\vdots$ | $\vdots$ | $\vdots$ |
|  | $n$ | $\mathbf{x}_n$ | $y_n$ | $\widehat{\mathbf{y}}_n = (\widehat{y}_{1,n}, \ldots, \widehat{y}_{K,n})^{\mathrm{T}}$ |
| Unlabeled data $\mathcal{U}$ | $n+1$ | $\mathbf{x}_{n+1}$ | ? | $\widehat{\mathbf{y}}_{n+1} = (\widehat{y}_{1,n+1}, \ldots, \widehat{y}_{K,n+1})^{\mathrm{T}}$ |
|  | $\vdots$ | $\vdots$ | $\vdots$ | $\vdots$ |
|  | $N$ | $\mathbf{x}_N$ | ? | $\widehat{\mathbf{y}}_N = (\widehat{y}_{1,N}, \ldots, \widehat{y}_{K,N})^{\mathrm{T}}$ |

**Overarching goal**   The overarching goal of this paper is to propose a safe and adaptive algorithm that constructs valid and more informative inference for $\boldsymbol{\theta}^*$ by leveraging multiple predictions, regardless of their individual quality. Importantly, we do not require the predictions $\widehat{Y}_k$ to share the same magnitude or form either with each other or with $Y$. For example, when $Y$ is a binary label, each prediction $\widehat{Y}_k$ may be either categorical or continuous. We allow the generation process of $\widehat{Y}_k$ to be a black box, potentially depending not only on the observed feature $\mathbf{X}$ but also on some unobservable or latent variables.

## 3   KEY IDEA: ILLUSTRATION WITH MEAN ESTIMATION

We elaborate the intuition behind our approach using the example of mean estimation. Consider $\mathbf{s}(\mathbf{x}, y; \theta) = y - \theta$, which corresponds to the estimand $\theta^* = \mathbb{E}(Y)$, the outcome mean. The naive estimator is the averaged outcome of the labeled samples, $\widehat{\theta}^{\mathrm{nv}} = n^{-1} \sum_{i=1}^{n} y_i$. While this estimator is unbiased, it may suffer from high variance and mean squared error due to the limited size of the labeled dataset. To improve efficiency by leveraging the $K$ machine learning-predicted outcomes $\widehat{\mathbf{y}}_i$ from both labeled and unlabeled data, we introduce a family of unbiased estimators indexed by weights $\boldsymbol{\omega} = (\omega_1, \ldots, \omega_K)^{\mathrm{T}}$:

$$\widehat{\theta}(\boldsymbol{\omega}) := \frac{1}{n} \sum_{i=1}^{n} y_i + \sum_{k=1}^{K} \omega_k \left( \frac{1}{N-n} \sum_{i=n+1}^{N} \widehat{y}_{k,i} - \frac{1}{n} \sum_{i=1}^{n} \widehat{y}_{k,i} \right).$$

This family covers several existing methods as special cases. For example, when $\boldsymbol{\omega} = \mathbf{0}$, it reduced to the naive estimator $\widehat{\theta}^{\mathrm{nv}}$. When there is only one prediction, i.e. $K = 1$, and $\omega = 1$, it turns out to be the PPI estimator (Angelopoulos et al., 2023):

$$\widehat{\theta}^{\mathrm{ppi}} = \frac{1}{n} \sum_{i=1}^{n} y_i + \frac{1}{N-n} \sum_{i=n+1}^{N} \widehat{y}_i - \frac{1}{n} \sum_{i=1}^{n} \widehat{y}_i.$$

Assuming that $\mathrm{var}(\widehat{\mathbf{Y}})$ is positive definite. To find the *best* estimator among the family, we can compute the variance, equivalently, the mean squared error, of $\widehat{\theta}(\boldsymbol{\omega})$:

$$\mathbb{E}[\{\widehat{\theta}(\boldsymbol{\omega}) - \theta^*\}^2] = \underbrace{\frac{1}{n} \mathrm{var}(Y)}_{\text{naive estimator}} + \underbrace{\frac{N}{n(N-n)} \boldsymbol{\omega}^{\mathrm{T}} \mathrm{var}(\widehat{\mathbf{Y}}) \boldsymbol{\omega} - \frac{2}{n} \boldsymbol{\omega}^{\mathrm{T}} \mathrm{cov}(\widehat{\mathbf{Y}}, Y)}_{\text{additional term}}. \tag{3}$$

The first term of (3) is the variance of the naive estimator, while the second captures the variance contributed by leveraging ML predictions. Notice that (3) is a quadratic form of $\boldsymbol{\omega}$, which achieves the minimum at

$$\boldsymbol{\omega}^{\text{opt}} = \frac{N-n}{N} \operatorname{var}(\widehat{\mathbf{Y}})^{-1} \operatorname{cov}(\widehat{\mathbf{Y}}, Y). \tag{4}$$

The optimal weight $\boldsymbol{\omega}^{\text{opt}}$ is fully determined by and can be easily estimated from the available data. In practical applications, one can estimate $\boldsymbol{\omega}^{\text{opt}}$ as

$$\widehat{\boldsymbol{\omega}}^{\text{opt}} = \frac{N-n}{N} \left\{ \frac{1}{N} \sum_{i=1}^{N} (\widehat{\mathbf{y}}_i - \overline{\widehat{\mathbf{y}}})(\widehat{\mathbf{y}}_i - \overline{\widehat{\mathbf{y}}})^{\mathrm{T}} \right\}^{-1} \left\{ \frac{1}{n} \sum_{i=1}^{n} (\widehat{\mathbf{y}}_i - \overline{\widehat{\mathbf{y}}})(y_i - \overline{y}) \right\},$$

where $\overline{y} = n^{-1} \sum_{i=1}^{n} y_i$, and $\overline{\widehat{\mathbf{y}}} = N^{-1} \sum_{i=1}^{N} \widehat{\mathbf{y}}_i$.

In this paper, we refer to the resulting optimal estimator $\widehat{\theta}(\widehat{\boldsymbol{\omega}}^{\text{opt}})$ as the proposed SADA estimator, denoted as $\widehat{\theta}^{\text{sada}}$. In what follows, for certain theoretical analyses and illustrations, we do not distinguish between the estimator $\widehat{\theta}(\boldsymbol{\omega}^{\text{opt}})$ and the SADA estimator $\widehat{\theta}^{\text{sada}}$, as they are asymptotically equivalent. This simplification is made when it does not lead to confusion.

The SADA estimator $\widehat{\theta}^{\text{sada}}$ enjoys the following two attractive properties.

**Safety**    Based on (3) and (4), one can compute that

$$\mathbb{E}\{(\widehat{\theta}^{\text{sada}} - \theta^*)^2\} = \underbrace{\frac{1}{n} \operatorname{var}(Y)}_{\text{naive estimator}} - \underbrace{\left(\frac{1}{n} - \frac{1}{N}\right) \operatorname{cov}(\widehat{\mathbf{Y}}, Y)^{\mathrm{T}} \operatorname{var}(\widehat{\mathbf{Y}})^{-1} \operatorname{cov}(\widehat{\mathbf{Y}}, Y)}_{\text{efficiency gain}}. \tag{5}$$

The efficiency gain in (5) is always non-negative regardless of the quality of the predictions. It vanishes to zero if and only if $\operatorname{cov}(\widehat{\mathbf{Y}}, Y) = \mathbf{0}$, that means, none of the predictions are correlated with the ground truth $Y$. In this worst-case scenario, the optimal weight is automatically assigned to zero (i.e., $\boldsymbol{\omega} = \mathbf{0}$), reducing to the naive estimator. Except this worst-case scenario, the SADA estimator offers a guaranteed positive efficiency gain over the naive method and enables more informative inference for $\theta^*$ (see Theorem 1 and Appendix B for details in the general case).

**Adaptivity**    If any one of the predictions (no need to know which one) is highly accurate, the weight automatically picks it up and introduces an estimator with either a faster convergence rate or an improved estimation efficiency. We elaborate this in two scenarios.

First, consider the case where one of the predictions perfectly matches the ground truth, for example, $\widehat{Y}_1 \equiv Y$. As shown in Appendix C, the optimal weight in this case is $\boldsymbol{\omega}^{\text{opt}} = (1, 0, \dots, 0)^{\mathrm{T}} \cdot (N - n)/N$. This means that the algorithm selects the most accurate prediction, $\widehat{Y}_1$, for estimation while discarding the less informative ones. Then, the SADA estimator turns out to be $N^{-1} \sum_{i=1}^{N} \widehat{y}_{1,i}$, and $\mathbb{E}\{(\widehat{\theta}^{\text{sada}} - \theta^*)^2\} = N^{-1} \operatorname{var}(Y)$, same as the oracle estimator who knows the ground truth of the unlabeled data. In this case, the SADA estimator converges at a faster rate of $N^{-1/2}$.

Second, consider the restricted case where the predictions are deterministic functions of the available feature $\mathbf{X}$, i.e., $Y_k = \widehat{f}_k(\mathbf{X})$. Then the best prediction one can expect to fit the outcome is the conditional mean, $\mathbb{E}(Y \mid \mathbf{X})$, which minimizes the mean squared error between $Y$ and $f(\mathbf{X})$. Assume that $\widehat{Y}_1 \equiv \mathbb{E}(Y \mid \mathbf{X})$. As shown in Appendix C, the optimal weight in this case also equals $\boldsymbol{\omega}^{\text{opt}} = (1, 0, \dots, 0)^{\mathrm{T}} \cdot (N - n)/N$. We show in the following proposition that $\widehat{\theta}^{\text{sada}}$ achieves the semiparametric efficiency bound (see Appendix A for the definition).

**Proposition 1.** *Denote $r_i = 1$ for labeled units $i = 1, \dots, n$ and 0 for unlabeled units. Suppose $n/N \to \pi \in (0, 1)$ as $n, N \to \infty$. Assume $\widehat{Y}_k = \widehat{f}_k(\mathbf{X})$ for $k = 1, \dots, K$. Then the efficient influence function (EIF) for estimating $\theta^*$, based on labeled data $\mathcal{L} = \{(\mathbf{x}_i, y_i), i = 1, \dots, n\}$ and unlabeled data $\mathcal{U} = \{\mathbf{x}_i, i = n+1, \dots, N\}$, is $\phi_{\text{eif}}(r, \mathbf{x}, y) = r\pi^{-1}\{y - \mathbb{E}(Y \mid \mathbf{x})\} + \mathbb{E}(Y \mid \mathbf{x}) - \theta^*$. For the estimator $\widehat{\theta}^{\text{sada}}$, we have $\sqrt{N}(\widehat{\theta}^{\text{sada}} - \theta^*) \xrightarrow{d} \mathcal{N}\{0, \mathbb{E}(\phi_{\text{eif}}^2)\}$, which attains the semiparametric efficiency bound.*

*Remark 1* (Interpretation). We provide an intuitive interpretation of why the propose algorithm enjoys these properties from the perspective of projection. We can rewrite $\widehat{\theta}^{\text{sada}} = \widehat{\theta}^{\text{nv}} + N^{-1}\sum_{i=1}^{N} g(\widehat{\mathbf{y}}_i) - n^{-1}\sum_{i=1}^{n} g(\widehat{\mathbf{y}}_i)$, where $g(\widehat{\mathbf{y}}) = \widehat{\mathbf{y}}^{\text{T}}\boldsymbol{\omega}^{\text{opt}} = \widehat{\mathbf{y}}^{\text{T}}\text{var}(\widehat{\mathbf{Y}})^{-1}\text{cov}(\widehat{\mathbf{Y}}, Y)$ is the $L^2(\mathbb{P})$-projection of $Y$ on the space linearly spanned by $\widehat{\mathbf{Y}}$. The projection ensures the *safety* and *adaptivity* of the SADA estimator. Specifically, (i) when all predictions $\widehat{\mathbf{Y}}$ are inaccurate and uncorrelated with the ground truth $Y$, the projection $g(\widehat{\mathbf{y}})$ shrinks to zero, reducing the SADA estimator to the naive one–but never performing worse than it; and (ii) when some prediction, for example $\widehat{Y}_1$, most closely fits the ground truth $Y$, the projection of $Y$ to $\widehat{\mathbf{Y}}$ turns out to be $\widehat{Y}_1$, resulting in an estimator $\widehat{\theta}^{\text{sada}} = \widehat{\theta}^{\text{nv}} + N^{-1}\sum_{i=1}^{N} \widehat{y}_{1,i} - n^{-1}\sum_{i=1}^{n} \widehat{y}_{1,i}$. If additionally $\widehat{Y}_1 \equiv Y$, then $\widehat{\theta}^{\text{sada}} = N^{-1}\sum_{i=1}^{N} \widehat{y}_{1,i}$ achieves a faster convergence rate. Alternatively, if $\widehat{Y}_1 \equiv \mathbb{E}(Y \mid \mathbf{X})$, then $\widehat{\theta}^{\text{sada}}$ becomes the semiparametrically efficient estimator of $\mathbb{E}(Y)$.

*Remark 2* (Comparison with PPI++). In case of the mean estimation (or, more generally, a scalar parameter $\theta^*$) with $K = 1$, the SADA estimator $\widehat{\theta}^{\text{sada}}$ is equivalent to the PPI++ estimator (Angelopoulos et al., 2024). As shown later, for a vector-valued parameter, the PPI++ estimator is generally less efficient than the proposed algorithm. Moreover, the PPI++ approach cannot, in general, leverage multiple predictions ($K > 1$) simultaneously.

# 4 FULL PROTOCOL: SAFE AND ADAPTIVE AGGREGATION OF MULTIPLE PREDICTIONS

Considering the general parameter $\boldsymbol{\theta}^*$ defined in (1), we construct a family of unbiased estimators, $\widehat{\boldsymbol{\theta}}(\mathcal{W})$, indexed by the tuning parameters $\mathcal{W} = (\mathcal{W}_1^{\text{T}}, \mathcal{W}_2^{\text{T}}, \ldots, \mathcal{W}_K^{\text{T}})^{\text{T}} \in \mathbb{R}^{(Kp)\times p}$, which solves

$$\frac{1}{n}\sum_{i=1}^{n} \mathbf{s}(\mathbf{x}_i, y_i; \boldsymbol{\theta}) + \sum_{k=1}^{K} \mathcal{W}_k^{\text{T}}\left\{\frac{1}{N-n}\sum_{i=n+1}^{N} \mathbf{s}(\mathbf{x}_i, \widehat{y}_{k,i}; \boldsymbol{\theta}) - \frac{1}{n}\sum_{i=1}^{n} \mathbf{s}(\mathbf{x}_i, \widehat{y}_{k,i}; \boldsymbol{\theta})\right\} = \mathbf{0}. \quad (6)$$

Noting that the second term has zero expectation regardless of the choice of $\mathcal{W}$ or the performance of $\widehat{\mathbf{y}}$, therefore, equation (6) is always a feasible estimating equation for $\boldsymbol{\theta}^*$. In particular, the naive estimator $\widehat{\boldsymbol{\theta}}^{\text{nv}}$ is a special case with $\mathcal{W} = \mathbf{0}$. When there is only one prediction ($K = 1$): (i) if $\mathcal{W} = \mathbf{I}$, $\widehat{\boldsymbol{\theta}}(\mathcal{W})$ reduces to the PPI estimator (Angelopoulos et al., 2023); (ii) if $\mathcal{W} = \omega\mathbf{I}$ with the tuning parameter $\omega$ selected optimally, it reduces to the PPI++ estimator (Angelopoulos et al., 2024).

We propose to identify the *best* estimator by minimizing the mean squared error of $\widehat{\boldsymbol{\theta}}(\mathcal{W})$.

**Proposition 2.** *Among the family of estimators $\widehat{\boldsymbol{\theta}}(\mathcal{W})$, the optimal tuning parameter, $\mathcal{W}^{\text{opt}}$, that minimizes the mean squared error loss such that $\mathbb{E}[\{\widehat{\boldsymbol{\theta}}(\mathcal{W}^{\text{opt}}) - \boldsymbol{\theta}^*\}^{\otimes 2}] \preceq \mathbb{E}[\{\widehat{\boldsymbol{\theta}}(\mathcal{W}) - \boldsymbol{\theta}^*\}^{\otimes 2}]$ for any $\mathcal{W} \in \mathbb{R}^{(Kp)\times p}$ is*

$$\mathcal{W}^{\text{opt}} = \frac{N-n}{N}\text{var}\{\mathcal{S}(\mathbf{X}, \widehat{\mathbf{Y}}; \boldsymbol{\theta}^*)\}^{-1}\mathbb{E}\{\mathcal{S}(\mathbf{X}, \widehat{\mathbf{Y}}; \boldsymbol{\theta}^*)\mathbf{s}(\mathbf{X}, Y; \boldsymbol{\theta}^*)^{\text{T}}\},$$

*where $\mathcal{S}(\mathbf{x}, \widehat{\mathbf{y}}; \theta) := (\mathbf{s}(\mathbf{x}, \widehat{y}_1; \theta)^{\text{T}}, \cdots, \mathbf{s}(\mathbf{x}, \widehat{y}_K; \theta)^{\text{T}})^{\text{T}}$.*

The proof of Proposition 2 is given in Appendix D. Proposition 2 implies that $\widehat{\boldsymbol{\theta}}(\mathcal{W}^{\text{opt}})$ performs no worse than the naive estimator regardless of the quality of predictions. To clarify this, one can compute that $\mathbb{E}[\{\widehat{\boldsymbol{\theta}}(\mathcal{W}^{\text{opt}}) - \boldsymbol{\theta}^*\}^{\otimes 2}] = n^{-1}\mathbf{H}^{-1}\boldsymbol{\Sigma}_{\text{opt}}\mathbf{H}^{-1}$, where

$$\boldsymbol{\Sigma}_{\text{opt}} = \text{var}\{\mathbf{s}(\mathbf{X}, Y; \boldsymbol{\theta}^*)\} - \frac{N-n}{N}\mathbb{E}\{\mathbf{s}(\mathbf{X}, Y; \boldsymbol{\theta}^*)\mathcal{S}(\mathbf{X}, \widehat{\mathbf{Y}}; \boldsymbol{\theta}^*)^{\text{T}}\}$$
$$\times \text{var}\{\mathcal{S}(\mathbf{X}, \widehat{\mathbf{Y}}; \boldsymbol{\theta}^*)\}^{-1}\mathbb{E}\{\mathcal{S}(\mathbf{X}, \widehat{\mathbf{Y}}; \boldsymbol{\theta}^*)\mathbf{s}(\mathbf{X}, Y; \boldsymbol{\theta}^*)^{\text{T}}\}.$$

The matrix $\boldsymbol{\Sigma}_{\text{opt}}$ consists of two terms: the first term corresponds to the variance of the naive estimator, $\boldsymbol{\Sigma}_{\text{nv}}$, defined in (2), while the second term is a positive semi-definite matrix that represents an efficiency gain. Practically, one can estimate the optimal weight, $\mathcal{W}^{\text{opt}}$, by

$$\widehat{\mathcal{W}}^{\text{opt}} = \left\{\frac{1}{N}\sum_{i=1}^{N} \mathcal{S}(\mathbf{x}_i, \widehat{\mathbf{y}}_i; \widehat{\boldsymbol{\theta}})\mathcal{S}(\mathbf{x}_i, \widehat{\mathbf{y}}_i; \widehat{\boldsymbol{\theta}})^{\text{T}}\right\}^{-1}\left\{\frac{1}{n}\sum_{i=1}^{n} \mathcal{S}(\mathbf{x}_i, \widehat{\mathbf{y}}_i; \widehat{\boldsymbol{\theta}})\mathbf{s}(\mathbf{x}_i, y_i; \widehat{\boldsymbol{\theta}})^{\text{T}}\right\},$$

where $\widehat{\boldsymbol{\theta}}$ is some consistent estimator of $\boldsymbol{\theta}^*$, for example, $\widehat{\boldsymbol{\theta}}^{\mathrm{nv}}$. Then, the proposed SADA estimator is $\widehat{\boldsymbol{\theta}}^{\mathrm{sada}} = \widehat{\boldsymbol{\theta}}(\widehat{\mathcal{W}}^{\mathrm{opt}})$.

Before presenting the theoretical properties, we briefly summarize the whole protocol for calculating the proposed estimator $\widehat{\theta}^{\mathrm{sada}}$ below.

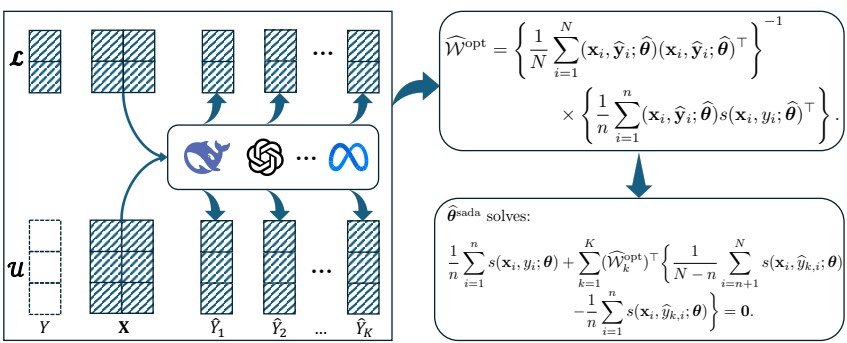

Figure 1: Protocol for computing the proposed SADA estimator $\widehat{\theta}^{\mathrm{sada}}$.

## 4.1 PROPERTIES OF THE SADA ESTIMATOR

We impose the following regularity assumptions.

**Assumption 1.** *(i) The parameter $\boldsymbol{\theta}^*$ lies in the interior of $\Theta$, and $\Theta$ is compact in $\mathbb{R}^p$; (ii) $\mathbb{E}\{\mathbf{s}(\mathbf{X}, Y; \boldsymbol{\theta})\} \neq 0$ if $\boldsymbol{\theta} \neq \boldsymbol{\theta}^*$; (iii) $\mathbf{s}(\mathbf{x}, y; \boldsymbol{\theta})$ is differentiable with respect to $\boldsymbol{\theta}$ in a neighborhood of $\boldsymbol{\theta}^*$, and $\mathbf{H} = \mathbb{E}\{\partial \mathbf{s}(\mathbf{X}, Y; \boldsymbol{\theta}^*)^{\mathrm{T}}/\partial \boldsymbol{\theta}\}$ exists and nonsingular. (iv) $\mathbf{s}(\mathbf{X}, Y; \boldsymbol{\theta})$ and $\mathbf{s}(\mathbf{X}, \widehat{Y}_k; \boldsymbol{\theta})$ have bounded first and second order moments in a neighborhood of $\boldsymbol{\theta}^*$ for $k = 1, \ldots, K$; (v) $\widehat{\mathcal{W}}^{\mathrm{opt}} \xrightarrow{p} \mathcal{W}^{\mathrm{opt}}$.*

**Theorem 1** (Safety). *Under Assumption 1, the SADA estimator $\widehat{\boldsymbol{\theta}}^{\mathrm{sada}}$ has the asymptotic representation $\sqrt{n}(\widehat{\boldsymbol{\theta}}^{\mathrm{sada}} - \boldsymbol{\theta}^*) \xrightarrow{d} \mathcal{N}(\mathbf{0}, \mathbf{H}^{-1}\boldsymbol{\Sigma}_{\mathrm{opt}}\mathbf{H}^{-1})$. More specifically, for its mean squared error, up to a negligible term, we have*

$$\mathbb{E}\{(\widehat{\boldsymbol{\theta}}^{\mathrm{sada}} - \boldsymbol{\theta}^*)^2\} = \frac{1}{n}\mathbf{H}^{-1}\boldsymbol{\Sigma}_{\mathrm{opt}}\mathbf{H}^{-1},$$

*where $\boldsymbol{\Sigma}_{\mathrm{opt}} = \boldsymbol{\Sigma}_{\mathrm{nv}} - (N - n)/N \cdot \boldsymbol{\Sigma}_g$, and*

$$\boldsymbol{\Sigma}_g = \mathbb{E}\{\mathbf{s}(\mathbf{X}, Y; \boldsymbol{\theta}^*)\mathcal{S}(\mathbf{X}, \widehat{\mathbf{Y}}; \boldsymbol{\theta}^*)^{\mathrm{T}}\}\operatorname{var}\{\mathcal{S}(\mathbf{X}, \widehat{\mathbf{Y}}; \boldsymbol{\theta}^*)\}^{-1}\mathbb{E}\{\mathcal{S}(\mathbf{X}, \widehat{\mathbf{Y}}; \boldsymbol{\theta}^*)\mathbf{s}(\mathbf{X}, Y; \boldsymbol{\theta}^*)^{\mathrm{T}}\}$$

*is always positive semi-definite. It implies that $\operatorname{var}(\widehat{\boldsymbol{\theta}}^{\mathrm{sada}}) \preceq \operatorname{var}(\widehat{\boldsymbol{\theta}}^{\mathrm{nv}})$.*

The proof of Theorem 1 is provided in Appendix E. In the general case, Theorem 1 shows that the SADA estimator enjoys a guaranteed efficiency gain over the naive estimator, regardless of the quality of $\widehat{\mathbf{Y}}$. This ensures valid and more informative inference for $\boldsymbol{\theta}^*$; see Appendix B for details. The following theorem formalizes the adaptivity property of the SADA estimator and is proved in Appendix F.

**Theorem 2** (Adaptivity). *(i) Suppose $\widehat{Y}_k \equiv Y$ for some $k$ and Assumption 1 holds, then we have $\boldsymbol{\Sigma}_g = \boldsymbol{\Sigma}_{\mathrm{nv}}$, and the SADA estimator $\widehat{\boldsymbol{\theta}}^{\mathrm{sada}}$ has the asymptotic representation $\sqrt{N}(\widehat{\boldsymbol{\theta}}^{\mathrm{sada}} - \boldsymbol{\theta}^*) \xrightarrow{d} \mathcal{N}(\mathbf{0}, \mathbf{H}^{-1}\boldsymbol{\Sigma}_{\mathrm{nv}}\mathbf{H}^{-1})$. More specifically, for its mean squared error, up to a negligible term, we have*

$$\mathbb{E}\{(\widehat{\boldsymbol{\theta}}^{\mathrm{sada}} - \boldsymbol{\theta}^*)^2\} = \frac{1}{N}\mathbf{H}^{-1}\boldsymbol{\Sigma}_{\mathrm{nv}}\mathbf{H}^{-1}.$$

*Note that this is the same as the oracle estimator who knows the ground truth of the unlabeled data; (ii) Suppose $n/N \to \pi \in (0, 1)$ as $n, N \to \infty$. Assume $\widehat{Y}_k = \widehat{f}_k(\mathbf{X})$ for $k = 1, \ldots, K$. Then the EIF for estimating $\boldsymbol{\theta}^*$, based on labeled data $\mathcal{L} = \{(\mathbf{x}_i, y_i), i = 1, \ldots, n\}$ and unlabeled data $\mathcal{U} = \{\mathbf{x}_i, i = n + 1, \ldots, N\}$, is*

$$\boldsymbol{\Phi}_{\mathrm{eif}}(r, \mathbf{x}, y) = -\mathbf{H}^{-1}\left[r\pi^{-1}\{\mathbf{s}(\mathbf{x}, y; \boldsymbol{\theta}^*) - \boldsymbol{\mu}(\mathbf{x})\} + \boldsymbol{\mu}(\mathbf{x})\right],$$

*where $\boldsymbol{\mu}(\mathbf{x}) = \mathbb{E}\{\mathbf{s}(\mathbf{x}, Y; \boldsymbol{\theta}^*) \mid \mathbf{x}\}$. Suppose $\mathbf{s}(\mathbf{x}, \widehat{y}_{k'}; \boldsymbol{\theta}^*) = \mathbf{s}(\mathbf{x}, \widehat{f}_{k'}(\mathbf{x}); \boldsymbol{\theta}^*) \equiv \boldsymbol{\mu}(\mathbf{x})$ for some $k'$, then we have $\sqrt{N}(\widehat{\boldsymbol{\theta}}^{\mathrm{sada}} - \boldsymbol{\theta}^*) \xrightarrow{d} \mathcal{N}\{\mathbf{0}, \mathbb{E}(\boldsymbol{\Phi}_{\mathrm{eif}} \boldsymbol{\Phi}_{\mathrm{eif}}^{\mathsf{T}})\}$, which attains the semiparametric efficiency bound.*

## 5 EXPERIMENTS

### 5.1 SYNTHETIC DATASETS

We first conduct small-scale simulations to evaluate the performance of the proposed algorithm. We generate $Y \sim N(\theta^*, 1)$, and generate two predictions $\widehat{Y}_1 = \gamma Y + (1 - \gamma)\epsilon_1$ and $\widehat{Y}_2 = (1 - \gamma)Y + \gamma\epsilon_2$, where $\gamma$ is a tuning parameter and $\epsilon_1, \epsilon_2$ are independent white noises drawn from the standard normal distribution. The true parameter $\theta^* = 0.5$. As $\gamma$ increases from 0 to 1, the quality of the prediction $\widehat{Y}_1$ improves in approximating the ground truth $Y$, while the prediction $\widehat{Y}_2$ becomes less accurate. We generate a dataset of size $N = 200$, among which $n = 60$ are labeled. We perform 1000 Monte Carlo replications.

Figure 2 reports the relative efficiency of different methods compared to the naive method, calculated as the ratio of their standard deviations across replications, as $\gamma$ increases from 0 to 1. Figure 2(a) shows that the performance of the PPI estimator heavily depends on the accuracy of the predictionit performs even worse than the naive estimator when the prediction is poor. Figure 2(b) demonstrates that PPI++ provides protection against poor prediction quality, yielding a variance that is never greater than that of the naive method. It reduces to the naive method when the prediction is noninformative. Figure 2(c) illustrates that our proposed SADA estimator not only consistently maintains a standard deviation lower than that of the naive estimator but also adaptively combines the strengths of $\widehat{Y}_1$ and $\widehat{Y}_2$ in their respective regions, regardless of which performs better, resulting in the most stable overall performance.

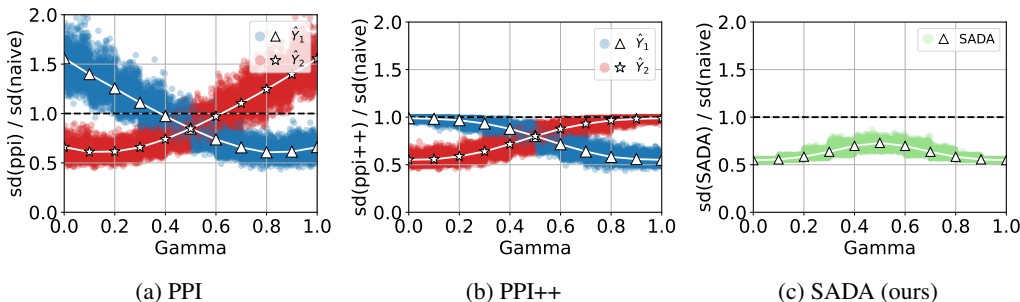

|  (a) PPI | (b) PPI++ | (c) SADA (ours) |

Figure 2: Relative efficiency of different methods compared to the naive method under varying prediction quality. Stars and triangles on the line indicate standard deviations over 1000 replications; scatter points represent those from individual replications.

### 5.2 BENCHMARK DATASETS

**Wine reviews** In this section, we apply our proposed methods to the wine review data published on WineEnthusiast: https://www.kaggle.com/datasets/mysarahmadbhat/wine-tasting. The dataset includes characteristics of the wine such as country, price, and region, along with reviews and ratings (from 80 to 100) from wine tasters. The goal is to estimate the mean rating. We use three predictive models–GPT-4o, Llama-3-8B, and DeepSeek–to generate predictions of the ratings based on the characteristics and the taster's reviews. Appendix G provides a step-by-step guidance of how to generate predictions using these LLMs. We sample 5000 instances for our analysis, using 1500 of them as unlabeled data and varying the number of labeled samples $n$ from 600 to 3500. The ratings of all 5000 instances are treated as the ground truth.

Figure 3 reports the variation of standard deviations of different methods as the labeled data size increases. The PPI and PPI++ methods in the subfigures are implemented using different single LLMs. While the standard deviations of all methods generally decrease with more labeled data, the

PPI estimator may perform worse than the naive method. In contrast, PPI++ guarantees improved efficiency over the naive approach, but the gains are limited when prediction accuracy is low, as seen in Figure 3(b). The SADA estimator, which integrates all predictions, consistently outperforms both PPI and PPI++ methods. Notably, its performance closely matches the performance of the best-performing PPI++ estimator using GPT-4o, highlighting its adaptive nature.

We also plot the oracle SADA estimator, SADAo, which includes the ground truth as one of its predictions. As shown, SADAo performs almost identically to the oracle estimator, calculated as the mean of the ground truth. This result verifies the desired adaptivity of the proposed method.

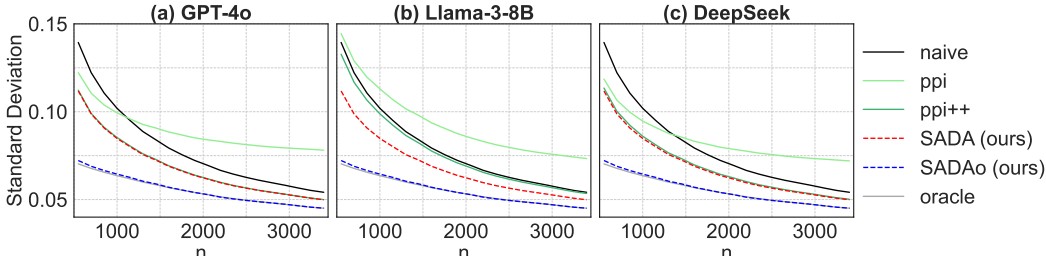

Figure 3: Comparison of standard deviations of different methods leveraging various prediction strategies. The estimand is the mean rating in the wine reviews dataset.

**Politeness of online requests** We also apply our method to study the relationship between politeness and the use of indicative modal features, using the dataset from Danescu-Niculescu-Mizil et al. (2013). From this dataset, we select 1000 requests from Stack Exchange and Wikipedia, each rated on a politeness scale from 1 to 25, averaged across five human annotators. The parameter of interest is the regression coefficient obtained by regressing the politeness score on the indicative modal features. We randomly designate 300 of the requests as unlabeled data and increase the number of labeled samples $n$ from 50 to 700.

Figure 4 shows similar results as Figure 3. In this task, the PPI estimator performs significantly worse than the naive method when predictions are inaccurate, as seen in Figure 4(a)-(b). Notably, PPI++ does not guarantee a lower variance than the naive approach in this setting. In contrast, SADA improves upon all other methods under different sample sizes.

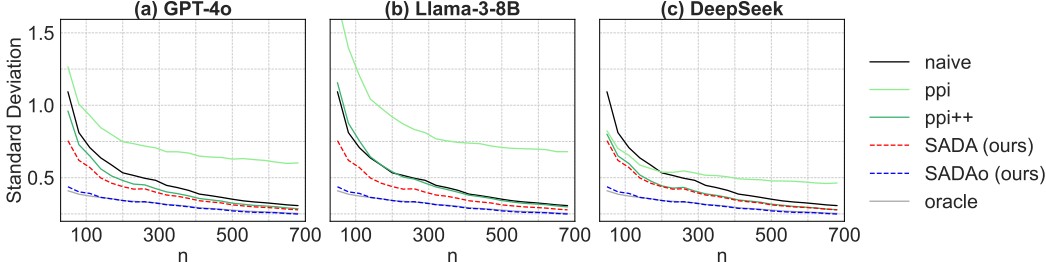

Figure 4: Comparison of standard deviations of different methods leveraging various prediction strategies. The estimand is the regression coefficient of politeness score on indicative modal features.

## 6 CONCLUSIONS

To wrap up, we propose a novel algorithm that safely and adaptively aggregates multiple predictions generated by different state-of-the-art ML models. Our method guarantees improved efficiency over the naive approach and demonstrates data adaptivity in balancing predictions of varying quality. Our method can be extended to the situations under distribution shift. In those settings, developing methods that are robust to distribution shift is essential for enhancing the reliability and practical effectiveness of semi-supervised learning.

## ETHICS STATEMENT

All authors have read and adhere to the ICLR Code of Ethics. Our work does not involve human subjects or sensitive data, and we have ensured compliance with privacy, fairness, and legal requirements. No conflicts of interest or harmful applications are involved.

## REPRODUCIBILITY STATEMENT

The authors ensure that all experimental details are provided in the main text or Appendix, and that the reported results are reproducible based on these details. The implementation code will be made publicly available on GitHub after the completion of the double-blind review process.

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

## A    SEMIPARAMETRIC EFFICIENCY BOUND

Here we briefly introduce the regular and asymptotically linear (RAL) estimator and influence function. In general, given i.i.d. copies of the random sample $\{\mathbf{z}_1, \ldots, \mathbf{z}_n\}$ with sample size $n$, an estimator for the parameter of interest $\boldsymbol{\beta}$, $\widehat{\boldsymbol{\beta}}$, is a RAL estimator if

$$\sqrt{n}(\widehat{\boldsymbol{\beta}} - \boldsymbol{\beta}) = n^{-1/2} \sum_{i=1}^{n} \boldsymbol{\phi}\left(\mathbf{z}_i\right) + o_p(1),$$

where the zero-mean function $\boldsymbol{\phi}(\cdot)$ is called the influence function of $\widehat{\boldsymbol{\beta}}$. Then, the central limit theorem implies that $\sqrt{n}(\widehat{\boldsymbol{\beta}} - \boldsymbol{\beta}) \xrightarrow{d} \mathcal{N}(\mathbf{0}, \mathbb{E}(\boldsymbol{\phi}\boldsymbol{\phi}^{\mathrm{T}}))$, provided that $\mathbb{E}(\boldsymbol{\phi}\boldsymbol{\phi}^{\mathrm{T}})$ is finite and nonsingular. Among all RAL estimators for $\boldsymbol{\beta}$, the influence function of the one with the smallest asymptotic variance is called the efficient influence function (EIF), $\boldsymbol{\phi}_{\mathrm{eif}}$, and the semiparametric efficiency bound is $\mathbb{E}\left(\boldsymbol{\phi}_{\mathrm{eif}}\boldsymbol{\phi}_{\mathrm{eif}}^{\mathrm{T}}\right)$.

We provide a brief, non-technical overview of the derivation of influence functions. At first, the likelihood can be decomposed into a parametric component and one or more nonparametric components. For each nonparametric component, we can identify a corresponding nuisance tangent space. The overall nuisance tangent space for the semiparametric model is then obtained by combining the individual nuisance tangent spaces. A valid influence function must lie in the orthogonal complement of this nuisance tangent space. Under suitable regularity conditions, an element from this perpendicular space can be chosen as the influence function.

We refer readers to Bickel et al. (1993) and Tsiatis (2006) for further interpretations. In addition, Kennedy (2023) provided several strategies for deriving the efficient influence function together with many examples.

## B    PROCEDURE OF INFERENCE

The asymptotic normality of $\widehat{\boldsymbol{\theta}}^{\mathrm{sada}}$ established in Theorem 1 enables us to construct confidence intervals for $\boldsymbol{\theta}^*$ following standard procedure. Specifically, let $\widehat{\mathbf{H}} = n^{-1} \sum_{i=1}^{n} \partial \mathbf{s}(\mathbf{X}_i, Y_i; \widehat{\boldsymbol{\theta}}^{\mathrm{sada}})/\partial\boldsymbol{\theta}$, $\widehat{\boldsymbol{\Sigma}}_{\mathrm{opt}} = \widehat{\boldsymbol{\Sigma}}_{\mathrm{nv}} - (N - n)/N \cdot \widehat{\boldsymbol{\Sigma}}_g$, where

$$\widehat{\boldsymbol{\Sigma}}_{\mathrm{nv}} = \frac{1}{n} \sum_{i=1}^{n} \mathbf{s}(\mathbf{X}_i, Y_i; \widehat{\boldsymbol{\theta}}^{\mathrm{sada}})\mathbf{s}(\mathbf{X}_i, Y_i; \widehat{\boldsymbol{\theta}}^{\mathrm{sada}})^{\mathrm{T}},$$

and

$$\widehat{\boldsymbol{\Sigma}}_g = \left\{ \frac{1}{n} \sum_{i=1}^{n} \mathbf{s}(\mathbf{X}_i, Y_i; \widehat{\boldsymbol{\theta}}^{\mathrm{sada}})\mathcal{S}(\mathbf{X}_i, \widehat{\mathbf{Y}}_i; \widehat{\boldsymbol{\theta}}^{\mathrm{sada}})^{\mathrm{T}} \right\}$$

$$\times \left\{ \frac{1}{N} \sum_{i=1}^{N} \mathcal{S}(\mathbf{X}_i, \widehat{\mathbf{Y}}_i; \widehat{\boldsymbol{\theta}}^{\mathrm{sada}})\mathcal{S}(\mathbf{X}_i, \widehat{\mathbf{Y}}_i; \widehat{\boldsymbol{\theta}}^{\mathrm{sada}})^{\mathrm{T}} \right\}^{-1}$$

$$\times \left\{ \frac{1}{n} \sum_{i=1}^{n} \mathcal{S}(\mathbf{X}_i, \widehat{\mathbf{Y}}_i; \widehat{\boldsymbol{\theta}}^{\mathrm{sada}})\mathbf{s}(\mathbf{X}_i, Y_i; \widehat{\boldsymbol{\theta}}^{\mathrm{sada}})^{\mathrm{T}} \right\}.$$

Let $\widehat{\boldsymbol{\Omega}} = \widehat{\mathbf{H}}^{-1}\widehat{\boldsymbol{\Sigma}}_{\mathrm{opt}}\widehat{\mathbf{H}}^{-1}$. Then, a confidence region $C_\alpha$ of $\boldsymbol{\theta}^*$ with error level $\alpha$ is given by

$$C_\alpha = \{\boldsymbol{\theta} \text{ such that } \sqrt{n}\widehat{\boldsymbol{\Omega}}^{-1/2}(\widehat{\boldsymbol{\theta}}^{\mathrm{sada}} - \boldsymbol{\theta}) \in \mathcal{R}_\alpha\},$$

where $\mathcal{R}_\alpha$ is a region in $\mathbb{R}^p$ such that $\mathbb{P}(\mathbf{Z}_p \in \mathcal{R}_\alpha) = 1 - \alpha$ for a $p$-dimensional standard normal random vector $\mathbf{Z}_p$. For point-wise inference of $j$th component of $\boldsymbol{\theta}^*$, $\theta_j^*$, let $\widehat{\boldsymbol{\Omega}}_{jj}$ be the $j$th diagonal entry of $\widehat{\boldsymbol{\Omega}}$. Then, a confidence interval $C_{j,\alpha}$ of $\boldsymbol{\theta}^*$ with error level $\alpha$ is given by

$$C_{j,\alpha} = \widehat{\theta}_j^{\mathrm{sada}} \pm \widehat{\boldsymbol{\Omega}}_{jj}^{1/2} Z_{1-\alpha/2}/\sqrt{n},$$

where $Z_{1-\alpha/2}$ is the $1 - \alpha/2$ upper quantile of the standard normal distribution.

## C    Proofs for the Mean Estimation

We provide the proofs for the two cases of the adaptivity properties described in Section 3.

**Proof of case 1 (the oracle case).**    Assume $\widehat{Y}_1 \equiv Y$. We denote $\widehat{\mathbf{Y}}_{-1} = (\widehat{\mathbf{Y}}_2, \ldots, \widehat{\mathbf{Y}}_K)^{\mathrm{T}}$, and

$$\mathrm{var}(\widehat{\mathbf{Y}}) = \begin{pmatrix} \mathrm{var}(Y) & \mathrm{cov}(\widehat{\mathbf{Y}}_{-1}, Y)^{\mathrm{T}} \\ \mathrm{cov}(\widehat{\mathbf{Y}}_{-1}, Y) & \mathrm{var}(\widehat{\mathbf{Y}}_{-1}) \end{pmatrix} =: \begin{pmatrix} C_{11} & \boldsymbol{C}_{21}^{\mathrm{T}} \\ \boldsymbol{C}_{21} & \boldsymbol{C}_{22} \end{pmatrix}.$$

By the inversion formula of $2 \times 2$ block matrix (Lu & Shiou, 2002, Theorem 2.1), we have

$$\mathrm{var}(\widehat{\mathbf{Y}})^{-1} = \begin{pmatrix} C_{11}^{-1} + C_{11}^{-1}\boldsymbol{C}_{21}\boldsymbol{M}^{-1}\boldsymbol{C}_{21}^{\mathrm{T}}C_{11}^{-1} & -C_{11}^{-1}\boldsymbol{C}_{21}\boldsymbol{M}^{-1} \\ -\boldsymbol{M}^{-1}\boldsymbol{C}_{21}^{\mathrm{T}}C_{11}^{-1} & \boldsymbol{M}^{-1} \end{pmatrix}, \tag{7}$$

and $\mathrm{cov}(\widehat{\mathbf{Y}}, Y) = (C_{11}, \boldsymbol{C}_{21}^{\mathrm{T}})^{\mathrm{T}}$, where $\boldsymbol{M} = \boldsymbol{C}_{22} - \boldsymbol{C}_{21}^{\mathrm{T}}C_{11}^{-1}\boldsymbol{C}_{21}$. Then

$$\boldsymbol{\omega}^{\mathrm{opt}} = \frac{N - n}{N}\mathrm{var}(\widehat{\mathbf{Y}})^{-1}\mathrm{cov}(\widehat{\mathbf{Y}}, Y) = \frac{N - n}{N}(1, 0, \ldots, 0)^{\mathrm{T}}.$$

Thus, we have $\widehat{\theta}^{\mathrm{sada}} = N^{-1}\sum_{i=1}^{N}\widehat{y}_{1,i}$, and

$$\mathbb{E}\{(\widehat{\theta}^{\mathrm{sada}} - \theta^*)^2\} = \frac{1}{N}\mathrm{var}(Y).$$

**Proof of case 2 (Proposition 1).**    We derive the EIF for the general estimating equation (6) in the proof of Theorem 2(ii). Since the mean estimation is a special case of (6), we omit its proof here.

Assume $\widehat{Y}_k = \widehat{f}_k(\mathbf{X})$ for $k = 1, \ldots, K$, and $\widehat{Y}_1 \equiv \mathbb{E}(Y \mid \mathbf{X})$. Let $\widehat{\mathbf{f}}_{-1}(\mathbf{x}) = (\widehat{f}_2(\mathbf{x}), \ldots, \widehat{f}_K(\mathbf{x}))^{\mathrm{T}}$. Then, we have

$$\mathrm{var}(\widehat{\mathbf{Y}}) = \begin{pmatrix} \mathrm{var}\{\mathbb{E}(Y \mid \mathbf{X})\} & \mathrm{cov}\{\widehat{\mathbf{f}}_{-1}(\mathbf{X}), \mathbb{E}(Y \mid \mathbf{X})\}^{\mathrm{T}} \\ \mathrm{cov}\{\widehat{\mathbf{f}}_{-1}(\mathbf{X}), \mathbb{E}(Y \mid \mathbf{X})\} & \mathrm{var}\{\widehat{\mathbf{f}}_{-1}(\mathbf{X})\} \end{pmatrix} =: \begin{pmatrix} D_{11} & \boldsymbol{D}_{21}^{\mathrm{T}} \\ \boldsymbol{D}_{21} & \boldsymbol{D}_{22} \end{pmatrix},$$

and

$$\mathrm{cov}(\widehat{\mathbf{Y}}, Y) = \mathrm{cov}\left(\begin{pmatrix} \mathbb{E}(Y \mid \mathbf{X}) \\ \widehat{\mathbf{f}}_{-1}(\mathbf{X}) \end{pmatrix}, Y\right) = \mathrm{cov}\left(\begin{pmatrix} \mathbb{E}(Y \mid \mathbf{X}) \\ \widehat{\mathbf{f}}_{-1}(\mathbf{X}) \end{pmatrix}, \mathbb{E}(Y \mid \mathbf{X})\right) =: \begin{pmatrix} D_{11} \\ \boldsymbol{D}_{21} \end{pmatrix}.$$

Similar to (7), by the inversion formula of $2 \times 2$ block matrix, we obtain

$$\boldsymbol{\omega}^{\mathrm{opt}} = \frac{N - n}{N}\mathrm{var}(\widehat{\mathbf{Y}})^{-1}\mathrm{cov}(\widehat{\mathbf{Y}}, Y) = \frac{N - n}{N}(1, 0, \ldots, 0)^{\mathrm{T}}.$$

Thus, we have $\widehat{\theta}^{\mathrm{sada}} = n^{-1}\sum_{i=1}^{n}\{y_i - \mathbb{E}(Y \mid \mathbf{x}_i)\} + N^{-1}\sum_{i=1}^{N}\mathbb{E}(Y \mid \mathbf{x}_i)$, and

$$\sqrt{N}(\widehat{\theta}^{\mathrm{sada}} - \theta^*) = \frac{\sqrt{N}}{n}\sum_{i=1}^{n}\{y_i - \mathbb{E}(Y \mid \mathbf{x}_i)\} + \frac{1}{\sqrt{N}}\sum_{i=1}^{N}\mathbb{E}(Y \mid \mathbf{x}_i)\} - \theta^*$$

$$= \frac{1}{\sqrt{N}}\sum_{i=1}^{N}\phi_{\mathrm{eif}}(r_i, \mathbf{x}_i, y_i) + o_p(1).$$

where $n/N \to \pi$ and

$$\phi_{\mathrm{eif}}(r, \mathbf{x}, y) = \frac{r}{\pi}\{y - \mathbb{E}(Y \mid \mathbf{x})\} - \mathbb{E}(Y \mid \mathbf{x}) - \theta^*.$$

## D  PROOF OF PROPOSITION 2

Recall $\mathcal{W} = (\mathcal{W}_1^{\mathrm{T}}, \mathcal{W}_2^{\mathrm{T}}, \ldots, \mathcal{W}_K^{\mathrm{T}})^{\mathrm{T}} \in \mathbb{R}^{(Kp)\times p}$, $\mathcal{S}(\mathbf{x}, \widehat{\mathbf{y}}; \theta) = (\mathbf{s}(\mathbf{x}, \widehat{y}_1; \theta)^{\mathrm{T}}, \cdots, \mathbf{s}(\mathbf{x}, \widehat{y}_K; \theta)^{\mathrm{T}})^{\mathrm{T}}$, and $\widehat{\boldsymbol{\theta}}(\mathcal{W})$ solves

$$\frac{1}{n}\sum_{i=1}^{n}\mathbf{s}(\mathbf{x}_i, y_i; \boldsymbol{\theta}) + \mathcal{W}^{\mathrm{T}}\left\{\frac{1}{N-n}\sum_{i=n+1}^{N}\mathcal{S}(\mathbf{x}_i, \widehat{\mathbf{y}}_i; \boldsymbol{\theta}) - \frac{1}{n}\sum_{i=1}^{n}\mathcal{S}(\mathbf{x}_i, \widehat{\mathbf{y}}_i; \boldsymbol{\theta})\right\} = \mathbf{0}.$$

Then the standard Taylor expansion yields that

$$\mathbf{0} = \frac{1}{n}\sum_{i=1}^{n}\mathbf{s}(\mathbf{x}_i, y_i; \boldsymbol{\theta}^*) + \mathcal{W}^{\mathrm{T}}\left\{\frac{1}{N-n}\sum_{i=n+1}^{N}\mathcal{S}(\mathbf{x}_i, \widehat{\mathbf{y}}_i; \boldsymbol{\theta}^*) - \frac{1}{n}\sum_{i=1}^{n}\mathcal{S}(\mathbf{x}_i, \widehat{\mathbf{y}}_i; \boldsymbol{\theta}^*)\right\}$$

$$+ \left[\frac{1}{n}\sum_{i=1}^{n}\frac{\partial}{\partial\boldsymbol{\theta}^{\mathrm{T}}}\mathbf{s}(X, Y; \overline{\boldsymbol{\theta}}) + \mathcal{W}^{\mathrm{T}}\left\{\frac{1}{N-n}\sum_{i=n+1}^{N}\frac{\partial}{\partial\boldsymbol{\theta}^{\mathrm{T}}}\mathcal{S}(\mathbf{x}_i, \widehat{\mathbf{y}}_i; \overline{\boldsymbol{\theta}}) - \frac{1}{n}\sum_{i=1}^{n}\frac{\partial}{\partial\boldsymbol{\theta}^{\mathrm{T}}}\mathcal{S}(\mathbf{x}_i, \widehat{\mathbf{y}}_i; \overline{\boldsymbol{\theta}})\right\}\right]$$

$$\times \{\widehat{\boldsymbol{\theta}}(\mathcal{W}) - \boldsymbol{\theta}^*\},$$

where $\overline{\boldsymbol{\theta}}$ lies between $\widehat{\boldsymbol{\theta}}(\mathcal{W})$ and $\boldsymbol{\theta}^*$. Then following uniform weak law of large number (Newey & McFadden, 1994) under the regularity conditions, we have

$$\widehat{\boldsymbol{\theta}}(\mathcal{W}) - \boldsymbol{\theta}^* \doteq -\mathbf{H}^{-1}\left[\frac{1}{n}\sum_{i=1}^{n}\mathbf{s}(\mathbf{x}_i, y_i; \boldsymbol{\theta}^*) + \mathcal{W}^{\mathrm{T}}\left\{\frac{1}{N-n}\sum_{i=n+1}^{N}\mathcal{S}(\mathbf{x}_i, \widehat{\mathbf{y}}_i; \boldsymbol{\theta}^*) - \frac{1}{n}\sum_{i=1}^{n}\mathcal{S}(\mathbf{x}_i, \widehat{\mathbf{y}}_i; \boldsymbol{\theta}^*)\right\}\right],$$

where $\mathbf{H} = \mathbb{E}\{\partial\mathbf{s}(X, Y; \boldsymbol{\theta}^*)^{\mathrm{T}}/\partial\boldsymbol{\theta}\}$. Then

$$\mathbb{E}[\{\widehat{\boldsymbol{\theta}}(\mathcal{W}) - \boldsymbol{\theta}^*\}^{\otimes 2}] \doteq \mathbf{H}^{-1}\left[\frac{1}{n}\mathrm{var}\{\mathbf{s}(X, Y; \boldsymbol{\theta}^*)\} + \frac{N}{n(N-n)}\mathcal{W}^{\mathrm{T}}\mathrm{var}\{\mathcal{S}(X, \widehat{Y}; \boldsymbol{\theta}^*)\}\mathcal{W}\right.$$

$$\left. - \frac{1}{n}\mathbb{E}\{\mathbf{s}(X, Y; \boldsymbol{\theta}^*)\mathcal{S}(X, \widehat{Y}; \boldsymbol{\theta}^*)^{\mathrm{T}}\}\mathcal{W} - \frac{1}{n}\mathcal{W}^{\mathrm{T}}\mathbb{E}\{\mathcal{S}(X, \widehat{Y}; \boldsymbol{\theta}^*)\mathbf{s}(X, Y; \boldsymbol{\theta}^*)^{\mathrm{T}}\}\right]\mathbf{H}^{-1}.$$

Let

$$\mathcal{W}^{\mathrm{opt}} = \frac{N-n}{N}\mathrm{var}\{\mathcal{S}(X, \widehat{Y}; \boldsymbol{\theta}^*)\}^{-1}\mathbb{E}\{\mathcal{S}(X, \widehat{Y}; \boldsymbol{\theta}^*)\mathbf{s}(X, Y; \boldsymbol{\theta}^*)^{\mathrm{T}}\}.$$

We next show that

$$\mathbb{E}[\{\widehat{\boldsymbol{\theta}}(\mathcal{W}^{\mathrm{opt}}) - \boldsymbol{\theta}^*\}^{\otimes 2}] \preceq \mathbb{E}[\{\widehat{\boldsymbol{\theta}}(\mathcal{W}) - \boldsymbol{\theta}^*\}^{\otimes 2}], \quad \forall \mathcal{W} \in \mathbb{R}^{(Kp)\times p}, \tag{8}$$

i.e., $\mathbb{E}[\{\widehat{\boldsymbol{\theta}}(\mathcal{W}) - \boldsymbol{\theta}^*\}^{\otimes 2}] - \mathbb{E}[\{\widehat{\boldsymbol{\theta}}(\mathcal{W}^{\mathrm{opt}}) - \boldsymbol{\theta}^*\}^{\otimes 2}]$ is a positive semi-definite matrix for any $\mathcal{W} \in \mathbb{R}^{(Kp)\times p}$. Note that

$$\mathbb{E}[\{\widehat{\boldsymbol{\theta}}(\mathcal{W}^{\mathrm{opt}}) - \boldsymbol{\theta}^*\}^{\otimes 2}] = \mathbf{H}^{-1}\left[\frac{1}{n}\mathrm{var}\{\mathbf{s}(X, Y; \boldsymbol{\theta}^*)\} - \left(\frac{1}{n} - \frac{1}{N}\right)\mathbb{E}\{\mathbf{s}(X, Y; \boldsymbol{\theta}^*)\mathcal{S}(X, \widehat{Y}; \boldsymbol{\theta}^*)^{\mathrm{T}}\}\right.$$

$$\left. \times \mathrm{var}\{\mathcal{S}(X, \widehat{Y}; \boldsymbol{\theta}^*)\}^{-1}\mathbb{E}\{\mathcal{S}(X, \widehat{Y}; \boldsymbol{\theta}^*)\mathbf{s}(X, Y; \boldsymbol{\theta}^*)^{\mathrm{T}}\}\right]\mathbf{H}^{-1}.$$

For any non-zero vector $\mathbf{a} \in \mathbb{R}^p$, let $\widetilde{\mathbf{a}} = \mathbf{H}^{-1}\mathbf{a}$, then we have

$$\mathbf{a}^{\mathrm{T}}\left(\mathbb{E}[\{\widehat{\boldsymbol{\theta}}(\mathcal{W}) - \boldsymbol{\theta}^*\}^{\otimes 2}] - \mathbb{E}[\{\widehat{\boldsymbol{\theta}}(\mathcal{W}^{\mathrm{opt}}) - \boldsymbol{\theta}^*\}^{\otimes 2}]\right)\mathbf{a}$$

$$= \frac{N}{n(N-n)}\left(\mathcal{W}\widetilde{\mathbf{a}} - \frac{N-n}{N}\mathrm{var}\{\mathcal{S}(X, \widehat{Y}; \boldsymbol{\theta}^*)\}^{-1}\mathbb{E}\{\mathcal{S}(X, \widehat{Y}; \boldsymbol{\theta}^*)\mathbf{s}(X, Y; \boldsymbol{\theta}^*)^{\mathrm{T}}\}\right)^{\mathrm{T}}$$

$$\times \mathrm{var}\{\mathcal{S}(X, \widehat{Y}; \boldsymbol{\theta}^*)\}$$

$$\times \left(\mathcal{W}\widetilde{\mathbf{a}} - \frac{N-n}{N}\mathrm{var}\{\mathcal{S}(X, \widehat{Y}; \boldsymbol{\theta}^*)\}^{-1}\mathbb{E}\{\mathcal{S}(X, \widehat{Y}; \boldsymbol{\theta}^*)\mathbf{s}(X, Y; \boldsymbol{\theta}^*)^{\mathrm{T}}\}\right)$$

$$\geq 0.$$

Therefore, (8) holds by definition.

# E    PROOF OF THEOREM 1

Following arguments similar to those in the proof of Proposition 2 in Appendix D, under assumption 1, we have

$$\widehat{\boldsymbol{\theta}}^{\text{sada}} - \boldsymbol{\theta}^*$$

$$\doteq -\mathbf{H}^{-1}\left[\frac{1}{n}\sum_{i=1}^{n}\mathbf{s}(\mathbf{x}_i, y_i; \boldsymbol{\theta}^*) + (\widehat{\mathcal{W}}^{\text{opt}})^{\text{T}}\left\{\frac{1}{N-n}\sum_{i=n+1}^{N}\mathcal{S}(\mathbf{x}_i, \widehat{\mathbf{y}}_i; \boldsymbol{\theta}^*) - \frac{1}{n}\sum_{i=1}^{n}\mathcal{S}(\mathbf{x}_i, \widehat{\mathbf{y}}_i; \boldsymbol{\theta}^*)\right\}\right]$$

$$= -\mathbf{H}^{-1}\left[\frac{1}{n}\sum_{i=1}^{n}\mathbf{s}(\mathbf{x}_i, y_i; \boldsymbol{\theta}^*) + (\mathcal{W}^{\text{opt}})^{\text{T}}\left\{\frac{1}{N-n}\sum_{i=n+1}^{N}\mathcal{S}(\mathbf{x}_i, \widehat{\mathbf{y}}_i; \boldsymbol{\theta}^*) - \frac{1}{n}\sum_{i=1}^{n}\mathcal{S}(\mathbf{x}_i, \widehat{\mathbf{y}}_i; \boldsymbol{\theta}^*)\right\}\right]$$

$$- \mathbf{H}^{-1}(\widehat{\mathcal{W}}^{\text{opt}} - \mathcal{W}^{\text{opt}})^{\text{T}}\left\{\frac{1}{N-n}\sum_{i=n+1}^{N}\mathcal{S}(\mathbf{x}_i, \widehat{\mathbf{y}}_i; \boldsymbol{\theta}^*) - \frac{1}{n}\sum_{i=1}^{n}\mathcal{S}(\mathbf{x}_i, \widehat{\mathbf{y}}_i; \boldsymbol{\theta}^*)\right\}$$

$$=: T_1 + T_2, \tag{9}$$

where $\mathbf{H} = \mathbb{E}\{\partial\mathbf{s}(\mathbf{X}, Y; \boldsymbol{\theta}^*)^{\text{T}}/\partial\boldsymbol{\theta}\}$. We next show that $T_2 = o_p(n^{-1/2})$. Recall that

$$\mathcal{W}^{\text{opt}} = \frac{N-n}{N}\text{var}\{\mathcal{S}(\mathbf{X}, \widehat{\mathbf{Y}}; \boldsymbol{\theta}^*)\}^{-1}\mathbb{E}\{\mathcal{S}(\mathbf{X}, \widehat{\mathbf{Y}}; \boldsymbol{\theta}^*)\mathbf{s}(\mathbf{X}, Y; \boldsymbol{\theta}^*)^{\text{T}}\} =: \frac{N-n}{N}\overline{\mathcal{W}}.$$

Then,

$$\|T_2\| \leq \left\|\mathbf{H}^{-1}\right\| \times \frac{N-n}{N}\left\|\widehat{\overline{\mathcal{W}}} - \overline{\mathcal{W}}\right\| \times \left\|\frac{1}{N-n}\sum_{i=n+1}^{N}\mathcal{S}(\mathbf{x}_i, \widehat{\mathbf{y}}_i; \boldsymbol{\theta}^*) - \frac{1}{n}\sum_{i=1}^{n}\mathcal{S}(\mathbf{x}_i, \widehat{\mathbf{y}}_i; \boldsymbol{\theta}^*)\right\|$$

$$= O(1) \times o_p\left(\frac{N-n}{N}\right) \times O_p\left(\sqrt{\frac{1}{N-n} + \frac{1}{n}}\right) = o_p(n^{-1/2}),$$

where the first equality is by $\widehat{\mathcal{W}}^{\text{opt}} \xrightarrow{p} \mathcal{W}^{\text{opt}}$, Chebyshev's inequality and

$$\mathbb{E}\left\|\frac{1}{N-n}\sum_{i=n+1}^{N}\mathcal{S}(\mathbf{x}_i, \widehat{\mathbf{y}}_i; \boldsymbol{\theta}^*) - \frac{1}{n}\sum_{i=1}^{n}\mathcal{S}(\mathbf{x}_i, \widehat{\mathbf{y}}_i; \boldsymbol{\theta}^*)\right\|^2 = \left(\frac{1}{N-n} + \frac{1}{n}\right)\left\|\text{var}\{\mathcal{S}(\mathbf{X}, \widehat{\mathbf{Y}}; \boldsymbol{\theta}^*)\}\right\|.$$

Moreover,

$$\text{var}(T_1) = \mathbf{H}^{-1}\left[\frac{1}{n}\text{var}\{\mathbf{s}(\mathbf{x}_i, y_i; \boldsymbol{\theta}^*)\} + \left(\frac{1}{N-n} + \frac{1}{n}\right)(\mathcal{W}^{\text{opt}})^{\text{T}}\text{var}\{\mathcal{S}(\mathbf{X}, \widehat{\mathbf{Y}}; \boldsymbol{\theta}^*)\}\mathcal{W}^{\text{opt}}\right.$$

$$\left. - \frac{2}{n}\text{cov}\{\mathbf{s}(\mathbf{x}_i, y_i; \boldsymbol{\theta}^*), \mathcal{S}(\mathbf{X}, \widehat{\mathbf{Y}}; \boldsymbol{\theta}^*)\}\mathcal{W}^{\text{opt}}\right]\mathbf{H}^{-1}$$

$$= \mathbf{H}^{-1}\left\{\frac{1}{n}\boldsymbol{\Sigma}_{\text{nv}} - \frac{N-n}{Nn}\boldsymbol{\Sigma}_g\right\}\mathbf{H}^{-1},$$

where $\boldsymbol{\Sigma}_{\text{nv}} = \text{var}\{\mathbf{s}(\mathbf{x}_i, y_i; \boldsymbol{\theta}^*)\}$ and

$$\boldsymbol{\Sigma}_g = \mathbb{E}\{\mathbf{s}(\mathbf{X}, Y; \boldsymbol{\theta}^*)\mathcal{S}(\mathbf{X}, \widehat{\mathbf{Y}}; \boldsymbol{\theta}^*)^{\text{T}}\}\text{var}\{\mathcal{S}(\mathbf{X}, \widehat{\mathbf{Y}}; \boldsymbol{\theta}^*)\}^{-1}\mathbb{E}\{\mathcal{S}(\mathbf{X}, \widehat{\mathbf{Y}}; \boldsymbol{\theta}^*)\mathbf{s}(\mathbf{X}, Y; \boldsymbol{\theta}^*)^{\text{T}}\}.$$

By the central limit theorem and the Slutsky's theorem, we have

$$\sqrt{n}(\widehat{\boldsymbol{\theta}}^{\text{sada}} - \boldsymbol{\theta}^*) \xrightarrow{d} N\left(\mathbf{0}, \mathbf{H}^{-1}\{\boldsymbol{\Sigma}_{\text{nv}} - \frac{N-n}{N}\boldsymbol{\Sigma}_g\}\mathbf{H}^{-1}\right).$$

# F    PROOF OF THEOREM 2

**Proof of (i).**    Without loss of generality, we assume $\widehat{Y}_1 \equiv Y$. We denote $\mathbf{s} \equiv \mathbf{s}_1(\mathbf{X}, \widehat{Y}_1; \theta^*) \equiv \mathbf{s}(\mathbf{X}, Y; \theta^*)$ and $\mathbf{s}_{-1} \equiv (\mathbf{s}(\mathbf{X}, \widehat{Y}_2; \theta^*)^{\text{T}}, \cdots, \mathbf{s}(\mathbf{X}, \widehat{Y}_K; \theta^*)^{\text{T}})^{\text{T}}$. Then

$$\text{var}\{\mathcal{S}(\mathbf{X}, \widehat{\mathbf{Y}}; \theta^*)\} = \begin{pmatrix} \text{var}(\mathbf{s}_1) & \text{cov}(\mathbf{s}_{-1}, \mathbf{s}_1)^{\text{T}} \\ \text{cov}(\mathbf{s}_{-1}, \mathbf{s}_1) & \text{var}(\mathbf{s}_{-1}) \end{pmatrix},$$

and $\mathbb{E}\{\mathcal{S}(\mathbf{X},\widehat{\mathbf{Y}};\boldsymbol{\theta}^*)\mathbf{s}(\mathbf{X},Y;\boldsymbol{\theta}^*)^{\mathrm{T}}\} = (\mathrm{var}(\mathbf{s}_1),\mathrm{cov}(\mathbf{s}_{-1},\mathbf{s}_1)^{\mathrm{T}})^{\mathrm{T}}$. Similar to the proof in Appendix C, by the inversion formula of block matrix, we have $\mathcal{W}^{\mathrm{opt}} = (\mathbf{I},\mathbf{0},\dots,\mathbf{0})^{\mathrm{T}}\cdot(N-n)/N$, and

$$\boldsymbol{\Sigma}_g = \mathbb{E}\{\mathbf{s}(\mathbf{X},Y;\boldsymbol{\theta}^*)\mathcal{S}(\mathbf{X},\widehat{\mathbf{Y}};\boldsymbol{\theta}^*)^{\mathrm{T}}\}\,\mathrm{var}\{\mathcal{S}(\mathbf{X},\widehat{\mathbf{Y}};\boldsymbol{\theta}^*)\}^{-1}\mathbb{E}\{\mathcal{S}(\mathbf{X},\widehat{\mathbf{Y}};\boldsymbol{\theta}^*)\mathbf{s}(\mathbf{X},Y;\boldsymbol{\theta}^*)^{\mathrm{T}}\}$$
$$= \mathrm{var}(\mathbf{s}_1) = \boldsymbol{\Sigma}_{\mathrm{nv}}.$$

Then by Theorem 2, we have

$$\widehat{\boldsymbol{\theta}}^{\mathrm{sada}} - \boldsymbol{\theta}^* \xrightarrow{d} \mathcal{N}\left(\mathbf{0},\frac{1}{N}\mathbf{H}^{-1}\boldsymbol{\Sigma}_{\mathrm{nv}}\mathbf{H}^{-1}\right).$$

It is asymptotically equivalent to the oracle estimator who knows the ground truth of the unlabeled data and solves

$$\frac{1}{N}\sum_{i=1}^{N}\mathbf{s}(\mathbf{x}_i,y_i;\boldsymbol{\theta}) = 0.$$

**Proof of (ii).** We first derive the EIF for estimating $\boldsymbol{\theta}^*$, based on labeled data $\mathcal{L} = \{(\mathbf{x}_i,y_i), i = 1,\dots,n\}$ and unlabeled data $\mathcal{U} = \{\mathbf{x}_i, i = n+1,\dots,N\}$. The joint density from one observation is

$$\{\pi f(\mathbf{x})f(y\mid\mathbf{x})\}^r\{(1-\pi)f(\mathbf{x})\}^{1-r} = \pi^r(1-\pi)^{1-r}f(\mathbf{x})f(y\mid\mathbf{x})^r.$$

It's straightforward to show the tangent space of this model is $\mathcal{T} = \Lambda_1\bigoplus\Lambda_2$, where

$$\Lambda_1 = \{r\mathbf{b}(y,\mathbf{x}) : \mathbb{E}(\mathbf{b}\mid\mathbf{x}) = 0\},\quad\text{and}\quad\Lambda_2 = \{\mathbf{a}(\mathbf{x}) : \mathbb{E}(\mathbf{a}) = 0\}.$$

Then, by the orthogonal conditions (Tsiatis, 2006, Theorems 4.2 and 4.3), we can obtain

$$\boldsymbol{\Phi}_{\mathrm{eif}}(r,\mathbf{x},y) = -\mathbf{H}^{-1}\left[\frac{r}{\pi}\{\mathbf{s}(\mathbf{x},y;\boldsymbol{\theta}^*) - \boldsymbol{\mu}(\mathbf{x})\} + \boldsymbol{\mu}(\mathbf{x})\right],$$

where $\boldsymbol{\mu}(\mathbf{x}) = \mathbb{E}\{\mathbf{s}(\mathbf{x},Y;\boldsymbol{\theta}^*)\mid\mathbf{x}\}$, $\frac{r}{\pi}\{\mathbf{s}(\mathbf{x},y;\boldsymbol{\theta}^*) - \boldsymbol{\mu}(\mathbf{x})\}\in\Lambda_1$ and $\boldsymbol{\mu}(\mathbf{x})\in\Lambda_2$.

Without loss of generality, we assume $\mathbf{s}(\mathbf{x},\widehat{y}_1;\boldsymbol{\theta}^*) = \mathbf{s}(\mathbf{x},\widehat{f}_1(\mathbf{x});\boldsymbol{\theta}^*)\equiv\boldsymbol{\mu}(\mathbf{x})$. We denote $\mathbf{s}_{-1}(\mathbf{x}) = (\mathbf{s}(\mathbf{x},\widehat{f}_2(\mathbf{x});\boldsymbol{\theta}^*)^{\mathrm{T}},\dots,\mathbf{s}(\mathbf{x},\widehat{f}_K(\mathbf{x});\boldsymbol{\theta}^*)^{\mathrm{T}})^{\mathrm{T}}$. Then

$$\mathrm{var}\{\mathcal{S}(\mathbf{X},\widehat{\mathbf{Y}};\boldsymbol{\theta}^*)\} = \begin{pmatrix} \mathrm{var}\{\boldsymbol{\mu}(\mathbf{X})\} & \mathrm{cov}\{\mathbf{s}_{-1}(\mathbf{X}),\boldsymbol{\mu}(\mathbf{X})\}^{\mathrm{T}} \\ \mathrm{cov}\{\mathbf{s}_{-1}(\mathbf{X}),\boldsymbol{\mu}(\mathbf{X})\} & \mathrm{var}\{\mathbf{s}_{-1}(\mathbf{X})\} \end{pmatrix},$$

and $\mathbb{E}\{\mathcal{S}(\mathbf{X},\widehat{\mathbf{Y}};\boldsymbol{\theta}^*)\mathbf{s}(\mathbf{X},Y;\boldsymbol{\theta}^*)^{\mathrm{T}}\} = (\mathrm{var}\{\boldsymbol{\mu}(\mathbf{X})\},\mathrm{cov}\{\mathbf{s}_{-1}(\mathbf{X}),\boldsymbol{\mu}(\mathbf{X})\}^{\mathrm{T}})^{\mathrm{T}}$. Similar to the proof in Appendix C, by the inversion formula of block matrix,, we have $\mathcal{W}^{\mathrm{opt}} = (\mathbf{I},\mathbf{0},\dots,\mathbf{0})^{\mathrm{T}}\cdot(N-n)/N$. Then, by (9), we have

$$\sqrt{N}\{\widehat{\boldsymbol{\theta}}^{\mathrm{sada}} - \boldsymbol{\theta}^*\} = -\mathbf{H}^{-1}\left[\frac{1}{n}\sum_{i=1}^{n}\mathbf{s}(\mathbf{x}_i,y_i;\boldsymbol{\theta}^*) + \frac{N-n}{N}\left\{\frac{1}{N-n}\sum_{i=n+1}^{N}\mathbf{s}(\mathbf{x}_i,\widehat{y}_{1,i};\boldsymbol{\theta}^*) - \frac{1}{n}\sum_{i=1}^{n}\mathbf{s}(\mathbf{x}_i,\widehat{y}_{1,i};\boldsymbol{\theta}^*)\right\}\right]$$

$$= -\mathbf{H}^{-1}\left[\frac{\sqrt{N}}{n}\sum_{i=1}^{n}\{\mathbf{s}(\mathbf{x}_i,y_i;\boldsymbol{\theta}^*) - \boldsymbol{\mu}(\mathbf{x}_i)\} + \frac{1}{\sqrt{N}}\sum_{i=1}^{N}\widehat{\mu}(\mathbf{x}_i)\right] + o_p(1)$$

$$= \frac{1}{\sqrt{N}}\sum_{i=1}^{N}\boldsymbol{\Phi}_{\mathrm{eif}}(r_i,\mathbf{x}_i,y_i) + o_p(1),$$

due to $n/N\to\pi$.

## G  GENERATING PREDICTIONS USING LARGE LANGUAGE MODELS

This paper used large language models, including GPT-4o, Llamma-3-8B, and DeepSeek, to generate predictions as parts of the dataset in our empirical studies in Section 5. In this section, we provide step-by-step guidance on how these models were used.

The input to the LLM consists of four components: a *detail* section (providing natural language descriptions of each data column), a *background* (shared across all data points within a dataset), a *question* (same across all data points), and a list of *prompts* (10 in total, each reflecting a distinct tone

or inquiry style). Consequently, each data point yields 10 separate outputsone per promptand the final prediction is computed by averaging these outputs. Section G.1 and G.2 present example prompts used in the experiments of wine reviews and the politeness evaluation, respectively.

We recommend using at least 80 GB of RAM to run LLaMA-3-70B, though in our experiments, we used Llamma-3-8B as a substitute due to limited resources, which requires approximately 20 GB of RAM. For GPT-4o and DeepSeek-V3, we utilized API calls to improve efficiency and reduce infrastructure requirements.

The approximate data generation speeds for each model in our experiments are as follows: (i) Llama-3-8B: 1000 examples every 10 minutes; (ii) GPT-4o: 1000 examples every 100 minutes; and (iii) DeepSeek-V3: 1000 examples every 15 hours.

## G.1 EXAMPLE PROMPT FOR WINE REVIEWS

**Details:** The wine under review is called "Vermeil 2010 Luvisi Vineyard Zinfandel (Calistoga)". This wine has a price of $42.0/bottle of Zinfandel (wine variety) from Calistoga, Napa. It is produced by Vermeil and labeled as Luvisi Vineyard. The wine is described as: Gigantic, huge, enormously extracted Zinfandel. So strong, it's practically a food group. Offers tiers of raspberries, cherries, blackberries, black currants, raisins, dark chocolate and spices. The alcohol level is very high, giving it a hot finish. It's a classic example of this old vine, superripe style.

**Background:** You are a professional wine taster with expertise in U.S. wines. Based on wine descriptions, please evaluate the wine's quality and assign a score from 80 (lowest score) to 100 (highest score), just like a sommelier would.

**Question:** Choose an integer between 80 (lowest acceptable quality) and 100 (exceptional quality) to reflect your evaluation. No explanation is needed, and do not give any numbers other than the rating. Your answer must be in JSON format with an integer only, without additional text.

**Prompts:**

- Given the wine's description and background, how would you rate its overall quality?
- Considering the region, variety, and tasting notes, what score would you assign to this wine?
- Based on your expertise as a sommelier, where does this wine fall on the 80-100 quality scale?
- Reflecting on the sensory cues and price point, how would you evaluate this wine?
- Taking all available details into account, what would be a fair score for this wine?
- From a wine critic's perspective, how would you assess the quality of this U.S. wine?
- Using your tasting intuition and the provided features, what score best represents this wine's character?
- Judging from the varietal, region, and tasting profile, what rating would you give?
- As a seasoned wine evaluator, how would you numerically rate this wine's craftsmanship?
- Given the flavor profile and winemaking context, what score reflects your impression of this wine?

## G.2 EXAMPLE PROMPT FOR POLITENESS EVALUATION

**Details:** The text under review is: "Doing a redundant 'atoi' after you just did 'strtol' is probably the most twisted insult to the proper use of string-to-int conversion functions in C one can come up with. Why do you see the need to "re-convert" the value using the broken function 'atoi' when you already have it as 'val' from a proper function 'strtol'?" and is from the community: Stack Overflow.Please evaluate its level of politeness based on linguistic features.

**Background:** This is a post from an online blog. Based on its linguistic featuressuch as word choice and toneplease evaluate the level of politeness. Then, assign a score from 1(very impolite) to 25(extremely polite), just like a sommelier would.

**Question:** Choose an integer between 1 (lowest) and 25 (highest) to reflect your evaluation. No explanation is needed, and do not give any numbers other than the rating. Your answer must be in JSON format with an integer only, without additional text.

**Prompts:**

- Given the writing style and overall tone of this blog post, how would you rate its level of politeness?
- Taking into account the language used, the word choices, and the sentiment conveyed, what politeness score would you assign?
- Imagine you're an expert in online communication. Based on the phrasing and attitude in this post, what number would you give for politeness?
- As someone who understands tone and subtext, how polite does this blog post feel to you?
- Just like a sommelier tasting wine, how would you score the politeness of this post, based solely on its language?
- If you had to assign a politeness score between 1 and 25 to this writing sample, what would it be?
- What's your judgment on the tone of this blog posthow polite does it seem on a scale from 1 to 25?
- Considering the linguistic subtleties in this postformality, tone, and word choicehow would you evaluate its politeness?
- Reflecting on how this post might come across to a casual reader, what politeness score would you give it?
- Evaluate the post as if you were rating its etiquette. What score feels most appropriate?

