# OpenReview forum: "SADA: Safe and Adaptive Inference with Multiple Black-Box Predictions"
_ICLR.cc/2026/Conference — ICLR 2026 Conference Withdrawn Submission_

### Official Review · Reviewer_aYG4 · 2025-10-25

**Soundness:** 3
**Presentation:** 3
**Contribution:** 2
**Rating:** 4
**Confidence:** 4

**Summary:**

The paper proposes SADA (Safe and Adaptive Data Aggregation) — a novel framework for safe and adaptive semi-supervised inference that aggregates predictions from multiple black-box models (e.g., LLMs, deep networks) of unknown quality. The method aims to guarantee:
- Safety: Never performs worse (in mean squared error) than using labeled data alone — even if all predictions are poor.
- Adaptivity: If any one of the black-box predictions is highly accurate (without knowing which), SADA automatically leverages it to achieve semiparametric efficiency or a faster convergence rate.

SADA extends recent prediction-powered inference (PPI) work by Angelopoulos et al. (2023, 2024) to handle multiple predictors simultaneously, offering both theoretical guarantees and empirical validation.
Experiments on synthetic and real data (Wine reviews, Politeness datasets) show that SADA consistently outperforms naive, PPI, and PPI++ estimators — providing stable variance reduction and robust adaptation across scenarios.

**Strengths:**

- The paper is generally well-written. The intuition behind SADA is well-explained and intuitive. The connection to PPI and PPI++ is made explicit, situating SADA as a generalization.
- Theoretical contribution. Extends prediction-powered inference to multiple prediction sources under semi-supervised learning.
- Experiments on synthetic data and two real-world tasks demonstrates consistent improvement over PPI/PPI++ and robustness to poor prediction quality.

**Weaknesses:**

- Limited experimental depth. The benchmarks, while well-chosen, are relatively small-scale. There is no demonstration of SADA’s scalability to larger datasets and higher-dimensional parameters.
- Assumption realism.
It assumes $l_\theta(x; y)$ to be a convex loss function.
The assumption of having multiple predictions with overlapping but uncorrelated noise may not hold when using correlated LLMs (e.g., GPT-4 and GPT-4o-mini). Empirical tests on correlated predictors would strengthen the case.
- The estimation of optimal weight (line 323) depends on the whole set of data points (N), which is not feasible for large N.

**Questions:**

see above.

---

> ### Author Response · Authors · 2025-11-21
> **Response to Reviewer aYG4's Comments (1/2)**
>
> Dear Reviewer aYG4,
>
> We are sincerely grateful for your careful reading of our manuscript and the constructive feedback. Please find our point-by-point response to your comments below.
>
> > *[W1]. Limited experimental depth. The benchmarks, while well-chosen, are relatively small-scale. There is no demonstration of SADA’s scalability to larger datasets and higher-dimensional parameters.*
>
> **Response:**
>
> - Thank you for this question. To further demonstrate SADA’s scalability to more complicated task, we conduct an additional experiment on image classification.
> - **Dataset:** The experiment is conducted on the ImageNet-5 dataset, which consists of a random subset of five classes from ImageNet-1k, with 1300 images per class. Each class is split into 1000 training images, 150 validation images, and 150 test images, resulting in a total of 5000 training, 750 validation, and 750 test images across all classes. We randomly select 2500 images as the labeled data, with the remaining treated as unlabeled. To investigate the effect of labeled data quantity, we vary the proportion of labeled samples used in training from 0.2 (500 samples) to 1.0 (2500 samples) of the labeled set.
> - **Predictions:** We generate pseudo labels by training two teacher models, `EfficientNet-B0` and `RegNetY-008`, using the hyperparameters: 10 training epochs, a learning rate of 1e-3, a batch size of 256, and the AdamW optimizer. These trained models are then used to generate pseudo labels for 5000 training images.
> - **Baselines:** For comparison, we consider four baselines: (i) `labeled-only`, which trains solely on the labeled data; (ii) `pseudo-only`, which trains using pseudo labels generated for all training samples; (iii) `Mixture`, which uses ground-truth labels for labeled data and pseudo labels for unlabeled data; (iv) `DR Self-training`, which adopts the recently proposed doubly robust self-training method [68, NeurIPS 2023]. The baselines (ii)-(iv) use the pseudo-labels generated by EfficientNet-B0.
> - **Training:** We evaluate different methods by training `DaViT-T` (Ding et al., 2022, ECAV), a popular vision transformer architecture with state-of-the-art performance on ImageNet. The model is trained for 15 epochs using a learning rate of 1e-3, a batch size of 256, and the default AdamW optimizer.
> - **Results:** We report the top-1 accuracy of different methods with increasing fraction of labeled data in **Table 1**. It is shown that: (i) The SADA estimators consistently outperform all baselines. For example, SADA achieves about a 7.1% improvement over the doubly robust self-training method and about an 11.2% gain over the labeled-only method. (ii) Both the pseudo-only and mixture approaches perform worse than the labeled-only baseline, likely due to the unsatisfactory quality of pseudo labels. This indicates that directly incorporating pseudo labels may have a detrimental effect on performance.
>
> #### Table 1: Top-1 Accuracy on ImageNet-5.
>
> |Labeled fraction|0.2|0.4|0.6|0.8|1.0|
> |--|--|--|--|--|--|
> |Labeled-only|43.5%|50.0%|51.3%|52.9%|56.4%|
> |Pseudo-only|34.1%|36.4%|34.9%|34.5%|36.8%|
> |Mixture|34.5%|37.1%|41.1%|41.3%|45.3%|
> |DR Self-training|44.1%|47.9%|51.1%|54.9%|56.9%|
> |SADA (ours)|44.9%|52.1%|53.5%|58.8%|61.2%|
>
> *(to be continued, due to the space limit)*

---

> ### Author Response · Authors · 2025-11-21
> **Response to Reviewer aYG4's Comments (2/2)**
>
> > *[W2]. Assumption realism. It assumes $l_\theta(x,y)$ to be a convex loss function. The assumption of having multiple predictions with overlapping but uncorrelated noise may not hold when using correlated LLMs (e.g., GPT-4 and GPT-4o-mini). Empirical tests on correlated predictors would strengthen the case.*
>
> **Response:**
>
> - ***Convex loss function.***
>     - A convex loss is a standard condition in M-estimation and is widely used to ensure existence and uniqueness of $\theta^\star$, differentiability, and a well-behaved Hessian. It covers many commonly used parameters and models, including mean estimation, generalized linear models, and a wide range of convex risk minimization problems.
>     - As a slight extension, the target parameter $\theta^\star$ can also be defined based on a general estimating equation, i.e., $E[\mathbf{s}(X,Y;\theta^\star)] = 0$, under appropriate regularity conditions. In this setting, the proposed algorithm in (6) and all theoretical results continue to hold.
> - ***Correlated predictions.***  Regarding your comment on “overlapping but uncorrelated noise”, we would like to clarify that we actually do **NOT** impose any assumptions on the predictions themselves. Different predictions are allowed to be highly correlated with each other or nearly independent. These features are automatically handled by the optimal weight matrix. For illustration, consider equation (4) in the paper:
>     - If the predictions are approximately independent, then $\mathrm{var}(\hat{\mathbf{Y}})$ is close to a diagonal matrix, and its inverse mainly rescales each prediction to adjust for differences in magnitude.
>     -  If the predictions are highly correlated, on the other hand, $\mathrm{var}(\hat{\mathbf{Y}})$ becomes nearly singular along the redundant directions, and those redundant predictions are effectively downweighted. This mechanism applies when combining correlated LLMs such as GPT-4 and GPT-4o-mini.
>     - We'd like to report the correlation structure of the predictions in our experiments:
>         -  In the synthetic data, the correlation between $\hat y_1$ and $\hat y_2$ equals $\gamma(1-\gamma)$, which ranges from 0 to 0.25.
>         - In the wine review data, the correlation between labels predicted by GPT-4o and DeepSeek is 0.923, and the correlations between Llama-3-8B and GPT-4o and DeepSeek are 0.485 and 0.431, respectively.
>         - In the politeness data, the correlation between labels predicted by GPT-4o and DeepSeek is 0.819, and the correlations between Llama-3-8B and GPT-4o and DeepSeek are 0.290 and 0.277, respectively.
>         - It can be seen that, aAlthough GPT-4o and DeepSeek are highly correlated, the SADA estimator still stays robust and performs best.
>
>
>
>
> > *[W3]. The estimation of optimal weight (line 323) depends on the whole set of data points (N), which is not feasible for large N.*
>
> **Response:**
>
> - This is a great point of detail. In fact, our theory only requires **any consistent estimator** of the optimal weight matrix $\mathcal{W}^{\mathrm{opt}}$. We currently use all $N$ data points because $\hat{\mathbf{y}}$ is available for both labeled and unlabeled observations, and leveraging all $N$ samples can improve the estimation accuracy, especially when the labeled size $n$ is very small.
>
> - As you correctly points out, in some scenarios, for example when $N$ is extremely large or when pooling all data raises privacy or storage concerns, using all $N$ data point may be impractical. In such cases, it is totally valid to estimate the weight matrix using only the $n$ labeled data. This still yields a consistent estimator of $\mathcal{W}^{\mathrm{opt}}$, and all the asymptotic results continue to hold.
>
> - We will explicitly clarify this point in the revised version of the paper.

---

> ### Comment · Reviewer_aYG4 · 2025-11-27
> **Feedback to the authors' response**
>
> Convex loss function is indeed a limitation, while it is mainly used to ensure existence and uniqueness.
>
> > In such cases, it is totally valid to estimate the weight matrix using only the n labeled data.
>
> Apart from asymptotic results, it is not clear in practice what is the influence in reducing to use only the n labeled data.

---

> > ### Author Response · Authors · 2025-11-27
> > **Response to Reviewer aYG4's further comments**
> >
> > Dear Reviewer aYG4,
> >
> > Thank you for your feedback and for continuing the discussion with us. Please find our further responses to your comments below.
> >
> > > [Q1] Convex loss function is indeed a limitation, while it is mainly used to ensure existence and uniqueness.
> >
> > **Response:**
> > - We would like to clarify that, for the purpose of statistical inference, defining target parameter through a convex loss function $\ell_\theta$ is a very standard and common setup. It is known as the M-estimation [1, Section 5] and is widely used in the line of PPI literature and related works [2–4].
> > - As we noted in our original rebuttal, it is also valid to define the target parameter through general moment conditions/estimating equations, $\mathrm E[\mathbf{s}(X,Y;\theta^*)] = 0$. The parameter defined via a convex loss function is then a special case with $\mathbf{s}=\nabla_{\theta}\ell$.
> >
> >
> > > [Q2] Apart from asymptotic results, it is not clear in practice what is the influence in reducing to use only the n labeled data.
> >
> > **Response:**
> > - We would like to clarify that the specific method used to estimate the weights has very limited impact on the performance of the target parameter estimates, as long as the weight estimator is consistent.
> > - We double-checked our codes, and in our experiments the weights are estimated using only the labeled data. The results exhibit good finite-sample performance of the target parameter.
> >
> >
> > ### Reference
> > [1] Vaart, A. W. van der (1998) Asymptotic Statistics. Cambridge series in statistical and probabilistic mathematics. Cambridge: Cambridge University Press.
> > [2] Angelopoulos, A. N., Bates, S., Fannjiang, C., et al. (2023) Prediction-powered inference. *Science*, 382, 669–674.
> > [3] Angelopoulos, A. N., Duchi, J. C. and Zrnic, T. (2024) PPI++: Efficient prediction-powered inference. arXiv:2311.01453.
> > [4] Zhu, B., Ding, M., Jacobson, P., et al. (2023) Doubly-robust self-training. *NeurIPS*.

---

### Official Review · Reviewer_PHXa · 2025-10-27

**Soundness:** 4
**Presentation:** 3
**Contribution:** 3
**Rating:** 8
**Confidence:** 3

**Summary:**

This paper presents a method building on prediction-powered inference (PPI) to do unbiased inference when a small set of ground truth training labels and an assortment of synthetic prediction functions are available on unlabeled data. Theoretically and empirically, they demonstrate the strengths of this method as being no worse than a naive estimator, and being adaptively able to converge quickly to a good synthetic estimator if one exists.

**Strengths:**

- seems like a fairly clear and well-scoped contribution generalizing PPI++ to multiple predictors
- as far as I know it is novel, although someone who has more deeply read all of the PPI literature may disagre
- theory and experiments back up the general points made about the estimator

**Weaknesses:**

Not a ton of major weaknesses here - not ground breaking work but makes its point pretty cleanly afaict

- it is stated on L186 that predictions Y-hat do not need to have the same form either as each other or Y. The examples of categorical and continuous are given, but it's not clear how broadly this extends. I feels as though somehow they need to be operated on similarly but they can't be so different - for instance, Y-hat can't be a free-text output of an LLM it seems if Y is binary. Could use some clarification here
- In general, one useful baseline to look at here would be averaging the predictors and using that in PPI++ - this is a very natural thing to do when you have many predictors and would give a better sense I think of a strong ensemble-ish baseline
- Fig 2: Visually, we can see that the shape of the result curve in 2c is nice. It would be good to know, potentially in a table, if those results are actually better or worse than the individual PPI++ results
- would be good to run multiple seeds and show confidence bars in Fig 3/4, these experiments look promising but I want to know how significant the improvement is

small notes:
Line 36: should cite the LLMs (GPT, Llama, Deepseek)
L133: I get confused by column vector notations - clarify if this is the inner or outer product (the superscript cross product symbol you define)
Line 270: typo "propose" -> "proposed"
Fig 3: clarify that only the PPI lines differ between these subfigures, is that right? and the SADA lines are the same?

**Questions:**

What is the theoretical relationship to taking a weighted combination of the predictors as a function, using that in PPI++ and then optimizing the weights? This is not a high priority question but would help with some deeper understanding of the relationship of this method to PPI++

---

> ### Author Response · Authors · 2025-11-21
> **Response to Reviewer PHXa's Comments (1/3)**
>
> Dear Reviewer PHXa,
>
> We sincerely thank you for the highly positive evaluation and the thoughtful comments. Please find our point-by-point response to your comments.
>
>
> > [W1] It is stated on L186 that predictions Y-hat do not need to have the same form either as each other or Y. The examples of categorical and continuous are given, but it's not clear how broadly this extends. I feels as though somehow they need to be operated on similarly but they can't be so different - for instance, Y-hat can't be a free-text output of an LLM it seems if Y is binary. Could use some clarification here.
>
> **Response:**
> - Thank you for pointing out this question. The predition $\hat Y$ should ultimately be represented as a numerical quantity. What we mean by “different form” is that one prediction may be **continuous, categorical, or discrete-valued**, regardless of the form of $Y$ and other predictions. We will clarify this more clearly in the revision.
>
> > [W2] In general, one useful baseline to look at here would be averaging the predictors and using that in PPI++ - this is a very natural thing to do when you have many predictors and would give a better sense I think of a strong ensemble-ish baseline.
>
> **Response:**
> - Thank you for this very practical suggestion. Averaging the predictions and then applying PPI++ is indeed a natural ensemble-style baseline. However, we would like to clarify that simple averaging generally does **NOT** achieve the optimal use of the available predictions. To provide some intuition, we use synthetic-data to implement such an estimator and show results in Table 1. We observe that:
>     - When one prediction is poor (when $\gamma$ is close to 0 or 1), PPI++ using the averaged prediction performs poorly because the low-quality predictor reduces efficiency.
>     - When $\gamma = 0.5$, PPI++ with the averaged prediction performs the same as SADA, since in this symmetric case SADA’s optimal weight is exactly $(0.5, 0.5)$.
>
> - Therefore, to maximize efficiency, one should assign different weights to different predictions based on their qualities, which is exactly the underlying motivation of SADA.
>
> > [W3] Fig 2: Visually, we can see that the shape of the result curve in 2c is nice. It would be good to know, potentially in a table, if those results are actually better or worse than the individual PPI++ results.
>
> **Response:** Thank you for this helpful suggestion.  Table 1 below reports the exact numerical values corresponding to Figure 2. As shown in the table, SADA is consistently more efficient than both PPI and PPI++ across all other settings. We will add this table in the revised manuscript.
>
> #### Table 1: Relative efficiency of different methods compared to the naive baseline under varying prediction quality. Smaller values are better, and the best (or nearly best) value is hightlighted.
> |$\gamma$|0|0.1|0.2|0.3|0.4|0.5|0.6|0.7|0.8|0.9|1|
> |--|--|------|------|------|------|------|------|------|------|------|------|
> |SADA|**0.548**|**0.555**|**0.584**|**0.636**|**0.697**|**0.727**|**0.697**|**0.635**|**0.583**|**0.555**|**0.548**|
> |ppi ($\hat{y}_1$)|1.549|1.396|1.247|1.104|0.970|0.842|0.736|0.653|0.608|0.610|0.654|
> |ppi ($\hat{y}_2$)|0.655|0.609|0.611|0.656|0.737|0.845|0.968|1.103|1.245|1.392|1.551|
> |ppi++ ($\hat{y}_1$)|0.989|0.982|0.965|0.933|0.877|0.799|0.713|**0.636**|**0.582**|**0.556**|**0.549**|
> |ppi++ ($\hat{y}_2$)|**0.549**|**0.556**|**0.585**|0.639|0.715|0.801|0.876|0.932|0.965|0.982|0.989|
> |ppi++ (average)|0.798|0.776|0.757|0.742|0.731|0.727|0.728|0.740|0.756|0.776|0.799|
>
> *(to be continued, due to the space limit)*

---

> ### Author Response · Authors · 2025-11-21
> **Response to Reviewer PHXa's Comments (2/3)**
>
> > [W4] It would be good to run multiple seeds and show confidence bars in Fig 3/4, these experiments look promising but I want to know how significant the improvement is.
>
> **Response:**
> - Thank you for your comment. In response, we conducted 500 bootstrap resampling replicates to quantify the uncertainty of the standard deviation estimates reported in Figures 3 and 4. The resulting 95% variation intervals for these standard deviations are reported in Tables 2 and 3 below.
> - SADA shows not only a relatively small standard deviation but also little variation in standard deviation across the 500 replications. We hope these results provide stronger evidence for the stable and efficient performance of SADA.
>
> #### Table 2: 2.5%–97.5% percentile range of standard deviations over 500 replications in wine data.
> |method|n=1000|n=1600|n=2200|n=2800|n=3400|
> |--|--|--|--|--|--|
> |naive|[0.095,0.103]|[0.076,0.081]|[0.065,0.068]|[0.057,0.060]|[0.053,0.055]|
> |SADA|[0.079,0.085]|[0.066,0.070]|[0.058,0.061]|[0.052,0.055]|[0.049,0.051]|
> |ppi(Gpt4o)|[0.095,0.100]|[0.086,0.091]|[0.081,0.086]|[0.078,0.082]|[0.076,0.081]|
> |ppi(Llamma3b)|[0.095,0.101]|[0.086,0.090]|[0.079,0.083]|[0.073,0.076]|[0.070,0.073]|
> |ppi(DeepSeek)|[0.092,0.097]|[0.082,0.086]|[0.077,0.080]|[0.073,0.077]|[0.071,0.075]|
> |ppi++(Gpt4o)|[0.080,0.086]|[0.067,0.071]|[0.058,0.061]|[0.052,0.055]|[0.049,0.051]|
> |ppi++(Llamma3b)|[0.088,0.095]|[0.073,0.078]|[0.063,0.067]|[0.056,0.059]|[0.052,0.054]|
> |ppi++(DeepSeek)|[0.081,0.087]|[0.068,0.072]|[0.059,0.062]|[0.053,0.055]|[0.049,0.051]|
> |oracle|[0.062,0.065]|[0.056,0.058]|[0.051,0.053]|[0.047,0.049]|[0.044,0.046]|
> |SADAo|[0.060,0.065]|[0.055,0.058]|[0.050,0.053]|[0.046,0.048]|[0.044,0.046]|
>
>
> #### Table 3: 2.5%–97.5% percentile range of standard deviations over 500 replications in politeness data.
> |method|n=50|n=200|n=350|n=500|n=650|
> |--|--|--|--|--|--|
> |naive|[0.885,1.292]|[0.490,0.583]|[0.400,0.462]|[0.329,0.374]|[0.296,0.332]|
> |SADA|[0.567,0.846]|[0.395,0.475]|[0.339,0.401]|[0.29,0.334]|[0.267,0.301]|
> |ppi(Gpt4o)|[1.024,1.479]|[0.684,0.811]|[0.625,0.741]|[0.572,0.692]|[0.553,0.648]|
> |ppi(Llamma3b)|[1.254,2.15]|[0.797,1.046]|[0.669,0.839]|[0.649,0.779]|[0.634,0.732]|
> |ppi(DeepSeek)|[0.712,0.934]|[0.486,0.590]|[0.465,0.569]|[0.425,0.535]|[0.413,0.511]|
> |ppi++(Gpt4o)|[0.741,1.159]|[0.432,0.523]|[0.356,0.421]|[0.303,0.345]|[0.274,0.311]|
> |ppi++(Llamma3b)|[0.848,1.363]|[0.482,0.587]|[0.384,0.454]|[0.319,0.364]|[0.288,0.325]|
> |ppi++(DeepSeek)|[0.654,0.922]|[0.401,0.480]|[0.348,0.408]|[0.296,0.338]|[0.267,0.303]|
> |oracle|[0.383,0.443]|[0.323,0.363]|[0.293,0.325]|[0.262,0.29]|[0.245,0.269]|
> |SADAo|[0.331,0.518]|[0.309,0.376]|[0.282,0.332]|[0.253,0.288]|[0.239,0.269]|
>
> > [W5] small notes:
> > Line 36: should cite the LLMs (GPT, Llama, Deepseek)  L133: I get confused by column vector notations - clarify if this is the inner or outer product (the superscript cross product symbol you define)  Line 270: typo "propose" -> "proposed" Fig 3: clarify that only the PPI lines differ between these subfigures, is that right? and the SADA lines are the same?
>
> **Response:** Thank you for pointing out these typos. We will correct them in the revision. Regarding your second question, the notation $u^{\otimes 2}$ denotes the outer product of $u$ with itself. In Figure 3/4, only the PPI and PPI++ curves differ across the three panels. We will clarify these more explicitly in the revision.
>
> *(to be continued, due to the space limit)*

---

> ### Author Response · Authors · 2025-11-21
> **Response to Reviewer PHXa's Comments (3/3)**
>
> > [Q1] What is the theoretical relationship to taking a weighted combination of the predictors as a function, using that in PPI++ and then optimizing the weights? This is not a high priority question but would help with some deeper understanding of the relationship of this method to PPI++.
>
> **Response:** This is a great question. To formulate the problem, let $\hat{y_{\alpha}}=\sum_{k=1}^K \alpha_k \hat{y_k}$, $\sum_{k=1}^K \alpha_k=1$. If our understanding is correct, your proposal is to first form $\hat y_\alpha$, then apply PPI++ based on $\hat y_\alpha$ by jointly optimizing over $\lambda$ and $\alpha$.
> - In special cases such as mean estimation, this procedure is conceptually equivalent to SADA. Indeed, the augmented term in PPI++ is
> \begin{eqnarray*}
> \lambda\left[|\mathcal{U}|^{-1}\sum_{i\in\mathcal{U}}\hat y_{\alpha,i}-|\mathcal{L}|^{-1}\sum_{i\in\mathcal{L}}\hat y_{\alpha,i}\right]
> =\sum_{k=1}^K(\lambda\alpha_k)\left[|\mathcal{U}|^{-1}\sum_{i\in\mathcal{U}}\hat y_{k,i}-|\mathcal{L}|^{-1}\sum_{i\in\mathcal{L}}\hat y_{k,i}\right],
> \end{eqnarray*}
> which matches the SADA augmentation if we view $\lambda\alpha_k$ as our weight $w_k$.
> - In models such as least square estimation, this averaged PPI++ approach corresponds to a restricted sub-class of our estimator family in equation (6), and is therefore in general less efficient than SADA. Concretely, let $s(x,y;\theta)= y-x^T\theta$, then using $\hat y_\alpha$ leads to leads to an augmented term
> \begin{eqnarray*}
>  \lambda\left[|\mathcal{U}|^{-1}\sum_{i\in\mathcal{U}} s(x,\hat y_{\alpha,i};\theta)-|\mathcal{L}|^{-1}\sum_{i\in\mathcal{L}} s(x,\hat y_{\alpha,i};\theta)\right]
> = \sum_{k=1}^K(\lambda\alpha_k)\left[|\mathcal{U}|^{-1}\sum_{i\in\mathcal{U}} s(x,\hat y_{k,i};\theta)-|\mathcal{L}|^{-1}\sum_{i\in\mathcal{L}}s(x,\hat y_{k,i};\theta)\right],
> \end{eqnarray*}
> which is a special case of SADA class with $\mathcal{W}_k=\lambda\alpha_k\cdot I_p$.
> - For more general models, such as quantile regression, the two approaches could not be directly compared. However, the PPI++ strategy can be computationally challenging, because when $s$ is nonlinear in y, the dependence of $s(x,y;\theta)$ on $\alpha$ can make the joint optimization over $\lambda$ and $\alpha$ difficult. In contrast, the asymptotic variance of SADA is always a quadratic form in the weights, which yields an explicit optimal weight formula and a simple implementation.
> - We hope this discussion clarifies how SADA relates to, but differs from, the PPI++ approach and its variants. Once again, we appreciate your insightful comment, and will add a discussion about this in our revision.

---

> ### Comment · Reviewer_PHXa · 2025-11-26
> **Response**
>
> Thanks for the rebuttal. I'm already at a fairly high score here so won't be moving, but will keep an eye on other reviewers feedback.

---

### Official Review · Reviewer_YeDm · 2025-10-30

**Soundness:** 3
**Presentation:** 2
**Contribution:** 2
**Rating:** 4
**Confidence:** 2

**Summary:**

This paper proposes an augmented M-estimation framework that leverages multiple auxiliary signals ("black-box predictions", possibly by pretrained general-purpose language models) to reduce asymptotic variance relative to using labeled data alone, and shows adaptivity under idealized conditions.
Technically, the core idea of adding mean-zero augmentation terms built from auxiliary signals to an unbiased estimator and choosing weights to minimize asymptotic variance is classical, with prediction-powered inference (PPI) and its variants as the closest modern instantiation.
The main value appears to be packaging these tools for multi-surrogate, vector-parameter settings and clarifying efficiency guarantees.
The proposed method was evaluated on synthetic data and natural language data using LLMs.

**Strengths:**

- This paper proposed a general framework with clear recipe for augmenting M-estimators with multiple auxiliary signals, subsuming PPI as a special case.
- The author provided asymptotic variance/MSE dominance over the labeled-only estimator.
- It's proven that the proposed method achieves oracle efficiency if one auxiliary is ideal.
- The projection view makes the weighting understandable and implementable.

**Weaknesses:**

## Contribution

The author stated that

> First, we consider a semi-supervised setting where multiple sets of predicted labels are available, without making any assumptions about their quality or requiring prior knowledge of which predictions are more accurate. The predicted labels are also not needed to share the same scale or format, either with each other or with the true labels. This bridges the gap between advanced machine learning tools and principled methods for leveraging them to improve the inference results.

However, this statement is true for many other existing papers.
Conceptual novelty is unclear relative to classical augmented estimating equations or GMM weighting.
The authors should precisely articulate what is new (e.g., theory for multi-surrogate vector targets beyond existing results, sharper bounds?)

## Problem framing

> Given multiple predicted labels with unknown quality, how can we aggregate them in a safe and data-adaptive manner to improve estimation and inference?

The problem framing over-emphasizes "black-box predictions" and ignores the equivalence to multi-annotator or measurement-error settings; the difference from **crowdsourcing/ensembling** is not clarified.
Framing the auxiliaries as "black-box predictions" (and hinting they may be from LLMs) rather than "noisy annotators/surrogates" is a marketing choice, not a mathematical distinction.
Please clarify differences from crowdsourced label aggregation and model ensembling.

## "Safe"

> The proposed method is guaranteed to perform **no worse than** (?) the naive estimator (using the labeled data alone) in terms of mean squared error, regardless of the choice of machine learning models or their prediction accuracy.

The term "safe" is overloaded and used here to mean **asymptotic efficiency dominance** rather than robustness.
It seems several existing papers also use the term "safe" this way (https://arxiv.org/abs/2011.14185 PPI and this paper cited, and https://arxiv.org/abs/2502.17741), but in my opinion it's very confusing.
The paper should explain it with more precise language (e.g., asymptotically at least as efficient as labels-only, PSD variance dominance?)

## Writing

- "GPT, Llama, DeepSeek" should have references.
- The introduction makes broad claims ("bridges the gap between advanced machine learning tools and principled methods") without precise, falsifiable statements.

## Future work

> Our method can be extended to the situations under distribution shift. In those settings, developing methods that are robust to distribution shift is essential for enhancing the reliability and practical effectiveness of semi-supervised learning.

This is very generic, and I don't see a connection between the current work (safety?) and the distribution shift literature.
Either drop the generic sentence from the conclusion or make it more concrete (choose a shift model, transport the estimating equations with density-ratio weights, suggest a variance-dominance result, etc.).

**Questions:**

- The meaning of "gold-standard experiments" is unclear in the machine learning context.
- "Outputs from different models–such as GPT, Llama, or DeepSeek–often differ, sometimes substantial; and the quality of predictions from black-box models can be highly variable." Please provide empirical evidence or references.
- "In particular, low-quality or poorly calibrated predictions can introduce significant noise, increasing variance and leading to unreliable inference." How siginificant? How unreliable?
- What does "perfectly accurate" mean in this context?

---

> ### Author Response · Authors · 2025-11-21
> **Response to Reviewer YeDm's Comments (1/4)**
>
> Dear reviewer YeDm,
>
> We sincerely appreciate your careful reading of our manuscript and your constructive feedback. Below we provide a point-by-point response to your comments.
>
>
> > [W1] **(Contribution)**
> > However, this statement is true for many other existing papers. Conceptual novelty is unclear relative to classical augmented estimating equations or GMM weighting. The authors should precisely articulate what is new (e.g., theory for multi-surrogate vector targets beyond existing results, sharper bounds?)
>
> **Response:** Thank you for this insightful comments. We agree that semi-supervised settings with surrogate or noisy labels have been studied in various forms. This paper builds on the series of PPI literature and contributes a *general, data-driven, and provably safe* algorithm for the practically important case where multiple heterogeneous predictions are available.  Existing PPI methods and their extensions do not directly address this setting. Our main contributions and connection to augmented methods are as follows:
>
> -  **A principled method for ***multiple*** predictions.**
>     - Prior PPI/PPI++ and related works are primarily designed for access to a single prediction. In practice, with modern ML and LLMs, we routinely have many predictions that differ in form, scale, and quality. Existing methods do not extend trivially (see next bullet).
> In constrast, our paper introduces a principled approach that incorporates multiple predictions simultaneously without any model assumptions. This directly responds to a significant applied need.
>
> - **Adaptive and data-driven precedure of aggregation**
>     - SADA automatically identifies higher-quality predictions and assigns them greater weights, while downweighting poorer ones, leading to an effective aggregation. This yields a practical, user-friendly procedure for practitioners.
>     - As a comparison, if one wants to apply existing methods designed for a single prediction, one has to first compute the correlation of each prediction with the ground-truth outcome, select the best one, and then use only this prediction to train PPI or PPI++. This has two drawbacks:
> (i) It is not an efficient workflow for practitioners, especially when the number of predictions is large.
> (ii) It is statistically less efficient, since it uses information from only one prediction. This can be seen from our synthetic data experiments. Table 1 reports the exact numerical values corresponding to Figure 2 and shows more clearly that when $\gamma$ is around 0.5, SADA achieves significantly higher efficiency than PPI++ based on either prediction alone.
>
> - **A ***safety guarantee*** absent in existing approaches.**
>     - SADA provides a rigorous ***safety guarantee*** that the proposed estimator is never worse (in asymptotic MSE) than the labeled-only estimator, even when all predictions are arbitrarily poor or nearly pure noise. Existing methods like PPI or PPI++ generally donot guarantee this property. This can be seen in our experiments (such as Figure 4). This is crucial when predictions are from black-box systems where the quality of the pre-trained model is unknown or unclear.
>
> - **Connection to augmented estimating equation methods.**
>     - Our main idea behind SADA is to build an augmented estimating equation that incorporates multiple predictions. Instead of the classical approach that augments estimating equations using additional nuisance models, we construct a class of augmentation terms directly from the available predictions, which are unbiased for any predictions and weights. We then select the optimal one from this class by minimizing the asymptotic variance.
>     - When one prediction coincides with the ground truth labels, our estimating equation reduces to the estimating equation based on averaging over all samples, including both labeled and unlabeled data. The resulting estimator then attains a faster convergence rate, a property that classical augmented methods do not achieve.
>     - We also show in Theorem 2 that when one prediction equals the conditional mean, our estimating equation automatically coincides with the efficient influence function that classical augmented estimating equation methods typically aim to construct.
>
> #### Table 1: Relative efficiency of different methods compared to the naive method under varying prediction quality.
> |$\gamma$|0|0.1|0.2|0.3|0.4|0.5|0.6|0.7|0.8|0.9|1|
> |--|--|--|--|--|--|--|--|--|--|--|--|
> |SADA|0.548|0.555|0.584|0.636|0.697|0.727|0.697|0.635|0.583|0.555|0.548|
> |PPI++ ($\hat{y}_1$)|0.989|0.982|0.965|0.933|0.877|0.799|0.713|0.636|0.582|0.556|0.549|
> |PPI++ ($\hat{y}_2$)|0.549|0.556|0.585|0.639|0.715|0.801|0.876|0.932|0.965|0.982|0.989|
>
> *(to be continued, due to the space limit)*

---

> ### Author Response · Authors · 2025-11-21
> **Response to Reviewer YeDm's Comments (2/4)**
>
> > [W2] **(Problem framing)**
> > The problem framing over-emphasizes "black-box predictions" and ignores the equivalence to multi-annotator or measurement-error settings; the difference from crowdsourcing/ensembling is not clarified. Framing the auxiliaries as "black-box predictions" (and hinting they may be from LLMs) rather than "noisy annotators/surrogates" is a marketing choice, not a mathematical distinction. Please clarify differences from crowdsourced label aggregation and model ensembling.
>
> **Response:**
> - This is a great question that we like very much. We appreciate your expertise and knowledge about these relevant concepts including but not limited to, crowdsourced label aggregation, model ensembling, measurement error, multi annotator, noisy label, surrogate, etc. We have carefully examined and reviewed the relevant literature for this rebuttal. Below, we attempt to clarify the differences among them.
>
> - First of all, in our semi-supervised learning setting, there are two types of data (labeled and unlabeled). The first contains the high-quality **ground truth label** $y$, obtained through trusted but costly processes such as controlled experiments or expert annotation. These labels are accurate but limited. The labeled data are critical in defining the estimand and obtaining the labeled-data-only benchmark estimator. This setting is different from, say, crowdsourced label aggregation where the ground truth label is not available, e.g., \url{https://openreview.net/pdf?id=nFEQNYsjQO}.
>
> - The second type does not have labels; instead, it consists of multiple **predictions** $\hat{y}$, which can come from any low-cost source: crowdsourced annotators, weak-supervision pipelines, measurement-error surrogates, model ensembles, traditional ML models, or even foundation models where the quality of the pre-trained model is unknown or unclear. From this regard, the problem we consider shares certain similarity to the problems mentioned above, such as, crowdsourced label aggregation. However, their motivations and goals are different. Crowdsourcing label aggregation aims to combine noisy labels from multiple crowd workers to estimate a more accurate "true" label, by methods such as majority vote, averaging, or ensembling.
>
> - Instead, we place no structural assumptions on the quality, scale, or data-generating mechanism of these predictions. Mathematically, we regard the distribution of $Y$, $P_Y$, as the underlying distribution of labels that defines the target parameter $\theta^*$. In contrast, each prediction $\hat Y$ follows its own distribution $P_{\hat{Y}}$, which may be a noisy but related perturbation of $P_Y$, or may differ substantially from $P_Y$. Our framework does not impose any structural relationship between $P_Y$ and $P_{\hat{Y}}$. Our method provides a safe and adaptive way to aggregate any such predictions, regardless of how these predictions are produced. The goal is not to reconstruct a better label, but to safely and adaptively improve statistical estimation and inference using heterogeneous auxiliary predictions.
>
> - Once again, we think this is a great question. In the revision, we will devote a new subsection in the literature review to further clarify this point more explicitly.
>
>
> > [W3] **("Safe")**
> > The term "safe" is overloaded and used here to mean asymptotic efficiency dominance rather than robustness. It seems several existing papers also use the term "safe" this way (https://arxiv.org/abs/2011.14185 PPI and this paper cited, and https://arxiv.org/abs/2502.17741), but in my opinion it's very confusing. The paper should explain it with more precise language (e.g., asymptotically at least as efficient as labels-only, PSD variance dominance?)
>
> **Response:**
> - This is another great question. We like your insights very much! As you correctly note, “safe” in our paper means that SADA has no larger asymptotic variance than the labeled data only estimator, regardless of how the predictions perform. The rationale for this term is that, in statistical inference, the length and informativeness of confidence intervals are essentially determined by the asymptotic variance of the estimator. We therefore regard an estimator with larger variance as worse. In this sense, we call our procedure “safe” because it guarantees an asymptotic variance that is no larger than that of a naive estimator, and thus yields confidence intervals that are never worse than those from the naive approach.
> - We are considering to modify our title to "Efficiency-preserving adaptive inference with multiple black-box predictions" or "Performance-preserving adaptive inference with multiple black-box predictions" or something similar, and will avoid using "safe" without a clear articulation in the main text.
> - Regarding the termiology, if you have better ideas, please feel free to share with us. Once again, thank you very much!
>
> *(To be continued, due to the space limit)*

---

> ### Author Response · Authors · 2025-11-21
> **Response to Reviewer YeDm's Comments (3/4)**
>
> > [W4] **(Writing)**
> > - "GPT, Llama, DeepSeek" should have references.
> > - The introduction makes broad claims ("bridges the gap between advanced machine learning tools and principled methods") without precise, falsifiable statements.
>
> **Response:**
> - Thank you for this comment. we will add references for GPT, Llama, and DeepSeek in the revision.
> - Regarding the second point, our goal is to provide a principled procedure that uses powerful but opaque machine learned predictions to obtain provably valid and more efficient statistical inference. We will revise this sentence to state this more precisely and in a falsifiable way.
>
>
> > [W5] **(Future work)** This is very generic, and I don't see a connection between the current work (safety?) and the distribution shift literature. Either drop the generic sentence from the conclusion or make it more concrete (choose a shift model, transport the estimating equations with density-ratio weights, suggest a variance-dominance result, etc.).
>
>
> **Response:**
> - Thank you for pointing this out. We agree that distribution shift is a substantially different and far more challenging problem, and addressing it properly would require nontrivial methodological extensions. In particular, one would need to formally specify a shift model, modify the estimating equations accordingly, and develop new theory tailored to that specific form of shift. These steps go well beyond the current scope of our paper.
> - To avoid confusion, we will remove the generic sentence about distribution shift from the discussion section in the revision.
>
> > [Q1] The meaning of "gold-standard experiments" is unclear in the machine learning context.
>
> **Response:**  Thank you for this question. In this paper, “gold standard experiments” refers to trusted data collection procedures in which labels are produced through high quality but costly methods, such as expert assessment, careful human annotation, or controlled experimental measurement. These labels are treated as reliable ground truth. We will clarify this more clearly in the revision.
>
> > [Q2] "Outputs from different models–such as GPT, Llama, or DeepSeek–often differ, sometimes substantial; and the quality of predictions from black-box models can be highly variable." Please provide empirical evidence or references.
>
> **Response:** Thank you for this question. Our benchmark experiments give empirical evidence that model outputs vary in both scale and how well they match the ground truth.
> - In the wine data experiment, we use GPT-4o, Llama-3-8B, and DeepSeek to generate predicted ratings separately. The means of the ground truth and the three model predictions are 88.623, 89.863, 91.000, and 89.686, with standard deviations 3.155, 2.608, 1.891, and 2.324, respectively. The correlations between the three model predictions and the ground truth labels are 0.697, 0.307, and 0.679. The correlation between GPT-4o and DeepSeek predictions is 0.923, while the correlations between Llama-3-8B and GPT-4o and DeepSeek are 0.485 and 0.431.
> - In the politeness data experiment, the means of the ground truth and the three model predictions are 13.729, 16.235, 17.544, and 14.391, with standard deviations 3.612, 3.368, 1.767, and 3.048, respectively. The correlations between the model predictions and the ground truth labels are 0.697, 0.307, and 0.679. The correlation between GPT-4o and DeepSeek predictions is 0.819, and the correlations between Llama-3-8B and GPT-4o and DeepSeek are 0.290 and 0.277.
> - The data above clearly shows that the predictions from different models vary widely and differ in quality.
>
> *(to be continued, due to the space limit)*

---

> ### Author Response · Authors · 2025-11-21
> **Response to Reviewer YeDm's Comments (4/4)**
>
> > [Q3] "In particular, low-quality or poorly calibrated predictions can introduce significant noise, increasing variance and leading to unreliable inference." How siginificant? How unreliable?
>
> **Response:** Thank you for this question. There are two perspectives that clarify this point.
> - First, if one directly relies on a low-quality prediction, for example by treating it as the true label and pooling labeled and unlabeled data for inference, the resulting estimator can incur substantial bias, which then leads to poor coverage of the true parameter.
> - Second, if a low-quality prediction is used within methods such as PPI, the inference remains valid in terms of achieving nominal coverage. However, the variance of the estimator becomes larger than that of the naive baseline, which makes the confidence interval wider and the inference less informative. This point is shown in our Figures 2-4.
>
> > [Q4] What does "perfectly accurate" mean in this context?
>
> **Response:** The notion “perfectly accurate” refers to two concrete cases as discussed in the remark on adaptivity on page 5 (formally in Theorem 2):
> - When no restriction is imposed on the prediction $\hat Y$, perfect accuracy means $\hat Y = Y$, i.e., the ground truth label.
> - When we restrict the prediction to be a deterministic function of the features, $\hat Y = f(X)$, perfect accuracy corresponds to the conditional mean $\hat Y = E(Y \mid X)$, which is the best possible predictor under squared error loss.
> - In both cases, our theory shows that the optimal weight places value 1 on the perfectly accurate prediction and value 0 on all others, allowing the estimator to attain oracle performance.

---

### Official Review · Reviewer_Ecpf · 2025-10-31

**Soundness:** 3
**Presentation:** 3
**Contribution:** 1
**Rating:** 4
**Confidence:** 4

**Summary:**

This paper looks to leverage the availability of unlabelled data in order to improve statistical inference. Previous works have demonstrated that producing quality synthetic labels for these unlabelled data has become even easier with the proliferation of Large Language Models (LLMs). However, previous works leveraging unlabelled data and synthetic labels make use of only a single predictor. This work uses the predictions from several different models simultaneously to achieve strong variance reduction while maintaining consistency. Asymptotic guarantees are provided, as well as experiments on real and synthetic data to demonstrate the method’s efficacy.

**Strengths:**

- The problem being addressed is interesting and highly relevant to researchers and practitioners alike, both of which would be interested in better inference methods.
- The mathematical set up is well explained, and the annotations on equations help with building intuition.
- Several relevant baselines are compared with both theoretically and empirically, with the theoretical benefits of the new method being especially well demonstrated.
- In the mean estimation case (as well as others), using the the OLS estimator equivalent to equation (4) is an inspired way of ensuring that a perfectly predictive set of synthetic labels will receive a weight of 1, while other predictions will remain unweighted.
- The idea of leveraging multiple sets of predictions for the same inference problem is an interesting take on the typical PPI set up.

**Weaknesses:**

- [W1] The main weakness is that the method does not seem particularly different from PPI, PPI++, and stratified PPI. The same formula is used for finding the optimal weighting over the terms using predictions in the SADA estimator as for PPI++ in the K=1 case. Similarly, the idea of fitting multiple coefficients for multiple pools of data was explored in the stratified PPI paper, although there each of the coefficients are not fit simultaneously and not on data pools of the same size. Similarly, the guarantees around always performing better than the naive estimator are already guaranteed by PPI++, as already stated in this paper.
- [W2] Despite requiring several more sets of predictions than PPI++, SADA does not seem to be able to exceed the performance of PPI++ on the most correlated set of predictions. This is on display in both Figure 2 and Figure 3. See [Q1]. I would have anticipated that by leveraging multiple pools of predictions simultaneously, we could produce performance better than the best PPI++ estimator.
- [W3] (Minor) There are typos in the work (line 28, or requires -> or require) (line 36, sometimes substantial -> sometimes substantially)

**Questions:**

- [Q1] It seems as if this method is equivalent to using PPI++ with the most correlated set of data. How would this compare with first testing which set of predictions is most correlated using the labelled dataset, and then running PPI++ with that most correlated set? Estimating this correlation is already part of the process of estimating the correlation coefficient, which is $\lambda$ in the PPI literature and $\omega$ in this paper.

---

> ### Author Response · Authors · 2025-11-21
> **Response to Reviewer Ecpf's Comments (1/2)**
>
> Dear Reviewer Ecpf,
>
> We sincerely appreciate the time and effort you have dedicated to reviewing our work and providing valuable feedback. Below, we provide a point-by-point response to your comments. In particular, we would like to clarify how SADA outperforms PPI++ and demonstrate its greater practical usefulness.
>
> > [W1] The main weakness is that the method does not seem particularly different from PPI, PPI++, and stratified PPI. The same formula is used for finding the optimal weighting over the terms using predictions in the SADA estimator as for PPI++ in the K=1 case. Similarly, the idea of fitting multiple coefficients for multiple pools of data was explored in the stratified PPI paper, although there each of the coefficients are not fit simultaneously and not on data pools of the same size. Similarly, the guarantees around always performing better than the naive estimator are already guaranteed by PPI++, as already stated in this paper.
>
> **Response:** Thank you for this comment. We would like to clarify that SADA differs meaningfully from existing methods and enjoys several attractive properties that the existing methods do not offer. Specifically,
>
> - **SADA guanrantees performance no worse than the naive estiamtor, while PPI++ does not, even when $K=1$.**
> We would like to clarify that PPI++ only guarantees improvement over the naive method when the target parameter $\theta^*$ is a scaler, which in this case is conceptually equivalent to SADA as noted in Remark 2. For vector-valued parameters, such as regression coefficients, PPI++ no longer enjoys this guarantee, whereas SADA still does. The key reason is that SADA optimizes a full matrix weight, while PPI++ uses a single scalar weight. This seemingly small difference has an essential impact on efficiency and safety in the multivariate setting.
>
>
> - **SADA is designed to handle multiple predictions.**
> As a core motivation of this work, SADA is built for practical scenarios where multiple predictions from different sources are available. Existing methods such as PPI, PPI++, and stratified PPI, while insightful, cannot be directly applied in this multi-prediction setting. SADA therefore fills an important practical gap by providing an easy-to-implement procedure tailored for leveraging multiple predictions simultaneously.
>
>
> - **SADA performs better than PPI++ using the best single prediction.**
> Please see our response to `W2` for a detailed discussion. Intuitively, when there are two informative predictions, the latter strategy can incorporate information from only one of them at a time, whereas SADA can leverage both simultaneously by assigning them appropriate weights.
>
>
> - **SADA provides a user-friendly and fully data-driven procedure.**
> Even though in special cases (see our response to `W2`) where SADA is expected to perform similarly to PPI++ based on one best prediction, using PPI++ in practice requires extra steps: one has to first compute the correlation of each prediction with the ground-truth outcome, select the best one, and then use only this prediction to train PPI++. In contrast, SADA enables the use of all available predictions simtaneously, by automatically assigning higher weights to the more accurate ones. Thus, SADA offers a more efficient and user-friendly way to achieve the best possible performance, with guarantees that it is never worse than PPI++.
>
> - **SADA differs from stratified PPI in both the problem it addresses and the properties it offers.**
> Although the forms may look similar, stratified PPI targets heterogeneous domains and uses several weights to combine domain partitions, while SADA is designed to aggregate multiple predictions. Stratified PPI assumes predictions from a single autorater function $f:\mathcal{X}\to\mathbb{R}$ and does not apply to general predicted labels $\hat Y$ with different forms as we do. In addition, as with PPI++, stratified PPI guarantees improvement only relative to unstratified PPI; it does not ensure the no-worse-than-naive property that SADA offers.
>
> *(to be continued, due to the space limit)*

---

> > ### Author Response · Authors · 2025-11-21
> > **Response to Reviewer Ecpf's Comments (2/2)**
> >
> > > [W2] Despite requiring several more sets of predictions than PPI++, SADA does not seem to be able to exceed the performance of PPI++ on the most correlated set of predictions. This is on display in both Figure 2 and Figure 3. See [Q1]. I would have anticipated that by leveraging multiple pools of predictions simultaneously, we could produce performance better than the best PPI++ estimator.
> >
> > **Response:**
> > - Thank you for this comment. We would like to clarify that, leveraging multiple pools of predictions simultaneously, as SADA does, indeed yields performance that exceeds the best single-predictor PPI++ estimator in typical settings.
> >
> > - An intuitive explanation is as follows. Suppose we have two predictions of similar quality but based on different sources of information. PPI++ can effectively use only one of them at a time and therefore exploits only part of the available information. In contrast, SADA can simultaneously incorporate both predictions by assigning them appropriate weights, and thus more fully leverages the available information to improve efficiency.
> >
> > - We further illustrate this using synthetic data experiments. Table 1 reports the exact numerical values corresponding to Figure 2. It more clearly shows that when $\gamma$ is around 0.5 (i.e., $\hat y_1$ and $\hat y_2$ are of similar quality), SADA achieves substantially higher efficiency than PPI++ which is based on either prediction alone.
> >     - We also apologize that Figure 2 does not clearly show the exact values, which may give the impression that SADA performs nearly the same as PPI++ with the best prediction. In the revision, we will include a table with the exact values to make the comparison clearer.
> >
> > - We also acknowledge that, in some special cases, SADA is equivalent to PPI++. For example:
> >     - When $K=1$ and $p=1$, SADA is conceptually the same as PPI++;
> >     - When one of the predictions is perfectly accurate, SADA will ***automatically*** select this prediction by assigning a weight close to (1,0,...,0), and thus reduces to a PPI++ estimator. In this case, SADA is equivalent to first identifying the most correlated prediction and then using it in PPI++. This shows up in Table 1 for $\gamma$ near to 0 or 1.
> >     - In other cases, SADA is guaranteed to perform better than PPI++, since PPI++ is a particular element within our proposed class of estimators,  e.g., eq (6), and SADA is the optimal one in this class in the sense of minimizing variance/MSE.
> >     - Even so, SADA is a fully data-driven procedure so that practitioners do not need to manually select the best prediction and obtain an estimator that is never worse than PPI++.
> >
> > #### Table 1: Relative efficiency of different methods compared to the naive method under varying prediction quality.
> > |$\gamma$|0|0.1|0.2|0.3|0.4|0.5|0.6|0.7|0.8|0.9|1|
> > |--|--|------|------|------|------|------|------|------|------|------|------|
> > |SADA|0.548|0.555|0.584|0.636|0.697|0.727|0.697|0.635|0.583|0.555|0.548|
> > |PPI++ ($\hat{y}_1$)|0.989|0.982|0.965|0.933|0.877|0.799|0.713|0.636|0.582|0.556|0.549|
> > |PPI++ ($\hat{y}_2$)|0.549|0.556|0.585|0.639|0.715|0.801|0.876|0.932|0.965|0.982|0.989|
> >
> >
> >
> > > [W3] (Minor) There are typos in the work (line 28, or requires -> or require) (line 36, sometimes substantial -> sometimes substantially)
> >
> > **Response:** Thank you for pointing out these typos. We will correct them in the revision.
> >
> >
> >
> > > [Q1] It seems as if this method is equivalent to using PPI++ with the most correlated set of data. How would this compare with first testing which set of predictions is most correlated using the labelled dataset, and then running PPI++ with that most correlated set? Estimating this correlation is already part of the process of estimating the correlation coefficient, which is $\lambda$ in the PPI literature and $\omega$ in this paper.
> >
> > **Response:**
> > - Thank you for this insightful question. Please see our response to your `W2` for some details. In conclusion, this test-then-PPI++ strategy has some disadvantanges compared with SADA:
> > (i) In theory, the test-then-PPI++ procedure suffers from a double-dipping issue. This could potentially be addressed using sample splitting, but at the cost of reducing the effective sample size in the testing step as well as in the PPI++ step.
> > (ii) The test-then-PPI++ procedure ultimately relies on a single prediction; even the second most correlated prediction cannot be incorporated. In contrast, SADA can make use of all correlated predictions by automatically assigning them appropriate weights.
> > (iii) Another limitation of PPI++ is that it guarantees an efficiency gain only for the scalar parameter $\theta^*$. By contrast, SADA provides a **unified** framework with guarantees that handle a general **vector-valued** $\theta^*$ and **multiple** predictions.
> > - We hope this clarification helps to make our motivation and the usefulness of our method clearer.

---

### Note · Authors · 2025-12-03

I have read and agree with the venue's withdrawal policy on behalf of myself and my co-authors.